# Intracellular delivery of protein drugs with an autonomously lysing bacterial system reduces tumor growth and metastases

Vishnu Raman [1,2,7], Nele Van Dessel[1,2,7], Christopher L. Hall[1,2], Victoria E. Wetherby[2], Samantha A. Whitney[1], Emily L. Kolewe[1], Shoshana M. K. Bloom [1], Abhinav Sharma [1], Jeanne A. Hardy [3,4,5], Mathieu Bollen [6], Aleyde Van Eynde [6] & Neil S. Forbes [1,2,4,5 ✉]

Critical cancer pathways often cannot be targeted because of limited efficiency crossing cell membranes. Here we report the development of a Salmonella-based intracellular delivery system to address this challenge. We engineer genetic circuits that (1) activate the regulator *flhDC* to drive invasion and (2) induce lysis to release proteins into tumor cells. Released protein drugs diffuse from Salmonella containing vacuoles into the cellular cytoplasm where they interact with their therapeutic targets. Control of invasion with *flhDC* increases delivery over 500 times. The autonomous triggering of lysis after invasion makes the platform self-limiting and prevents drug release in healthy organs. Bacterial delivery of constitutively active caspase-3 blocks the growth of hepatocellular carcinoma and lung metastases, and increases survival in mice. This success in targeted killing of cancer cells provides critical evidence that this approach will be applicable to a wide range of protein drugs for the treatment of solid tumors.

[1] Department of Chemical Engineering, University of Massachusetts, Amherst, Amherst, MA, USA. [2] Ernest Pharmaceuticals, LLC, Hadley, MA, USA. [3] Department of Chemistry, University of Massachusetts, Amherst, Amherst, MA, USA. [4] Molecular and Cell Biology Program, University of Massachusetts, Amherst, Amherst, MA, USA. [5] Institute for Applied Life Science, University of Massachusetts, Amherst, Amherst, MA, USA. [6] Laboratory of Biosignaling & Therapeutics, Department of Cellular and Molecular Medicine, KU Leuven, Leuven, Belgium. [7] These authors contributed equally: Vishnu Raman, Nele Van Dessel. ✉email: forbes@umass.edu

Delivering protein drugs into the cytoplasm of cancer cells would expand the number of treatable cancer targets. More than 60% of the pathways that control cellular function are intracellular[1] and almost all are difficult to access. Intracellular pathways control most of the hallmarks of cancer[2] and have been the focus of a significant fraction of cancer research. Because of their specificity, protein biologics are excellent candidates for interfering with these pathways. However, bringing functional proteins across the cell membrane is technically challenging and current methods to deliver proteins intracellularly have poor efficacy[3–5]. Effective intracellular delivery, coupled with specific protein drugs, would provide treatments for previously incurable cancers.

Once engineered, intracellular bacteria can efficiently deliver proteins to tumors. Current delivery methods, including nanoparticles, cell-penetrating peptides, and antibody drug conjugates have limited efficacy because of (1) poor uptake into cells, (2) lack of specificity to cancer cells, (3) degradation in the circulatory system, and (4) failure to escape from endosomes[3,5–10]. When particles and proteins are taken up by cells, they are trafficked from early and late endosomes to lysosomes, where they are degraded[11–13]. A bacterial system would not be limited by these mechanisms.

Because of their physiology, Salmonella are uniquely suited to deliver proteins into cancer cells. Salmonella (1) actively invade cells[14,15], (2) specifically accumulate in tumors[16,17], (3) produce therapeutic molecules in situ[18,19], and (4) reshape endosomes into hospitable environments[14,20]. In the intestines, Salmonella evade host defense mechanisms by entering epithelial cells using proteins encoded by Salmonella pathogenicity island 1 (SPI1)[21–23]. This cellular invasion has not been measured in tumors. After invasion, the bacteria restructure endosomes into Salmonella-containing vacuoles (SCVs) by injecting proteins encoded by pathogenicity island 2 (SPI2)[24–26]. SCVs enable intracellular survival[24,27] and protect from intracellular defense mechanisms[28,29]. A critical step in the activation of SPI2 genes is the sensing of the endosomal environment, a mechanism that is unique to Salmonella.

A bacterial delivery system would have to incorporate three essential components: (1) synthesis of protein drugs, (2) invasion into cells, and (3) release of the drugs (Fig. 1a). Protein synthesis can be controlled with bacterial translation machinery[18,19]. Control of invasion is necessary to carry the produced proteins into cells. Invasion requires both flagella and the type III secretion system-1 (T3SS-1)[30,31]. Flagella are essential for cell invasion[30,31] because they sense the cell surface and determine the optimal location for invasion[32]. T3SS-1 is a needle apparatus that initiates invasion by injecting effector proteins into cells[33–35]. These proteins rearrange the actin cytoskeleton and induce endocytosis of the bacteria[36,37]. Production of these two bacterial structures is controlled by the factors *fliZ* and *hilD*, which are, in turn, controlled by the master regulator *flhDC*[38–41]. Protein release requires activation of Salmonella genes specifically inside cells. In Salmonella, this can be controlled with the promoters of SPI2 genes.

The use of bacteria changes what is traditionally meant by "delivery." Unlike traditional delivery vehicles, bacteria manufacture protein drugs at the disease site[42], delivering exponentially more molecules than were originally present in the injected bacteria. This specificity is the result of exponential growth in tumors, coupled with concurrent clearance from healthy organs[16,17,43–46]. Two potential protein drugs that exclusively affect intracellular proteins are the central domain of nuclear inhibitor of protein phosphatase 1 (NIPP1-CD) and constitutive two-chain active caspase-3 (CT Casp-3). NIPP1-CD induces cell death by competitively disrupting PP1 holoenzymes[47]. CT Casp-3 is an engineered form of caspase-3 that does not require intracellular activation. Caspase-3 is the dominant executioner caspase that causes apoptotic cell death[48,49].

Here, we describe the development of *Intracellular Delivering* (ID) *Salmonella*. We design this nonpathogenic, therapeutic strain to deliver proteins into solid tumors. It utilizes genetic circuits that control protein synthesis, invasion into cells, and release of protein drugs. This harnessing of the native invasion and survival machinery of Salmonella enables autonomous deposition of protein payloads directly into cancer cells. With this system, released proteins spread throughout the cytoplasm, are active, and interact with their cellular targets. In mouse models of breast cancer and hepatocellular carcinoma (HCC), ID Salmonella that produce CT Casp-3 are safe, decrease tumor growth and reduce established breast metastases. These outcomes show that this Salmonella-based delivery platform is an effective cancer therapy that renders the inside of cells more accessible to protein drugs.

## Results

**Intracellular lifestyle of Salmonella in tumors**. To determine the extent that Salmonella are intracellular in tumors, bacteria were administered to mice (Fig. 1b, c). A specialized strain of *Salmonella enterica* serovar Typhimurium was used that expresses a fluorescent reporter when intracellular (Supplementary Table 1). This strain was based on a therapeutic Salmonella strain (VNP20009; Δ*msbB*, Δ*purI*, Δ*xyl*) with an additional deletion (Δ*asd*) to enable plasmid retention in mice (Supplementary Table 1). In tumor-bearing mice, most Salmonella were intracellular and activated the reporter (black arrows; Fig. 1b, left), whereas some were exclusively extracellular (white arrows; Fig. 1b, right). Over all tumors, 70% of Salmonella were intracellular ($P < 0.0001$, $n = 5$, Fig. 1c).

The dependence of invasion on the regulator *flhDC* was determined by deleting *flhD* from the genome of the parental strain (Δ*flhD* Sal) and creating a strain with controllable re-expression (*flhDC* Sal; Supplementary Table 1). The parental Salmonella strain expresses native levels of *flhDC* and naturally invades into cancer cells in culture (arrows, Fig. 1d). Compared to uninduced control Salmonella (*flhDC*−), only bacteria with re-expressed *flhDC* (*flhDC*+; arrows) invaded into cells (Fig. 1e). Salmonella re-expressing *flhDC* invaded 84% of cells, which was 54 times greater than knockout controls ($P < 0.01$, Fig. 1f). In microfluidic tumor masses, which mimic tumor tissue bordering blood vessels, Salmonella with the intracellular GFP reporter and re-expressing *flhDC* invaded cells 53 times more than knockout controls ($P < 0.0001$, Fig. 1g). Salmonella re-expressing *flhDC* invaded cells throughout the tumor masses (Fig. 1g, bottom).

**Design of a protein release system**. Engineering Salmonella to release protein drugs required development of a system to trigger autonomous lysis after cell invasion (Fig. 2). This goal was achieved by identifying a SPI2 promoter that is triggered intracellularly and is not active extracellularly. Coupled to a GFP reporter, the promoters of two SPI2-associated genes, *PsseJ and PsifA*, both activated after invasion into cancer cells (white arrows; Fig. 2a, left). Despite both being activated intracellularly, the extracellular expression (black arrows) of *PsseJ* was 5.8 times less than *PsifA* ($P < 0.01$, Fig. 2a top), indicating that it is more sensitive to cell invasion. The activity of *PsseJ* increased more than four times after invasion ($P < 0.0001$, Fig. 2a bottom).

To release a synthesized protein cargo, the bacteria must lyse after invasion. Triggered expression of Lysin gene E (*LysE*) from bacteriophage ΦX1174 caused rapid bacterial death (Fig. 2b). After induction, Salmonella transformed with *PBAD-LysE* lysed

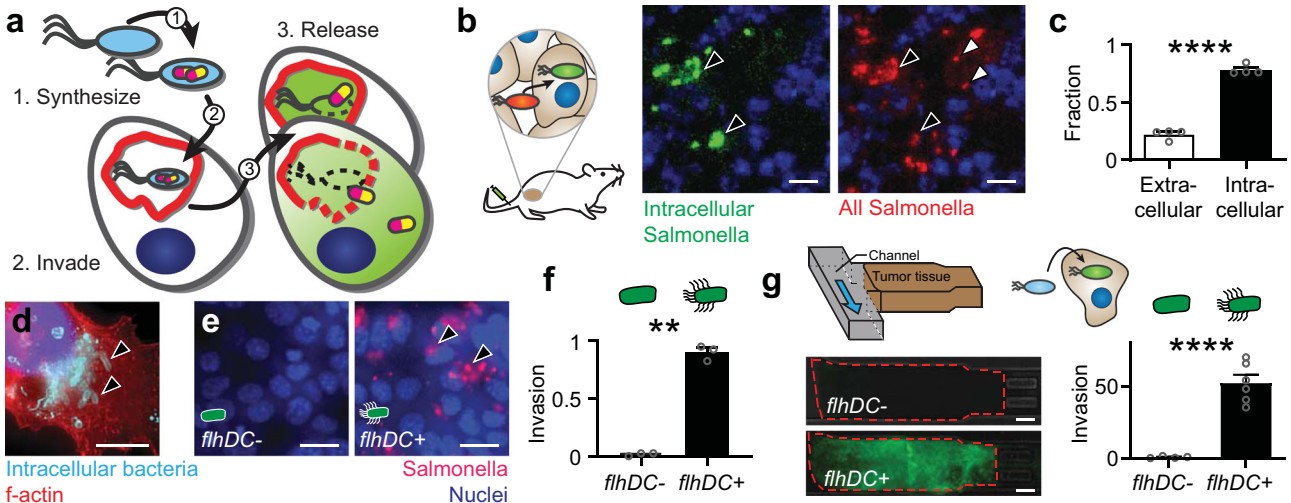

**Fig. 1 The intracellular lifestyle of Salmonella is controlled by *flhDC*. a** The design goals were to genetically engineer a bacterial vehicle that (1) synthesizes a protein drug (*yellow/purple*), (2) actively invades into cancer cells, and (3) releases drug, which escapes Salmonella vacuoles (SCVs, *red*). **b**, **c** Ninety-six hours after intratumoral injection of $2 \times 10^6$ CFU of intracellular-reporting Salmonella into subcutaneous 4T1 tumors in BALB/c mice, more bacteria (*red*) were intracellular (*green; black arrows*) than extracellular (*white arrows*; $P < 0.0001$; $n = 1258$ bacteria in 4 mice). **d**) In monolayer culture, Salmonella (*light blue, arrows*) invade cancer cells (*red*). Extracellular bacteria were removed with gentamicin (an invasion assay). **e** Knockout $\Delta flhD$ Salmonella were transformed with *PBAD-flhDC*. Uninduced bacteria (*flhDC-*) minimally invaded cancer cells. Induction with 20 mM arabinose (*flhDC+*) promoted invasion (*black arrows*). **f** Re-expression of *flhDC* significantly increased invasion ($P = 0.0012$; $n = 3$ independent biological samples). **g** In three-dimensional tumor-on-a-chip devices (*top left*), *flhDC+* Salmonella, with a green invasion reporter (*top right*), invaded more cells than *flhDC-* controls ($P < 0.0001$; $n = 6$ chambers for *flhDC+* and $n = 4$ for *flhDC-*). Data are shown as means ± SEM. Statistical comparisons in **c**, **f**, and **g**, and are two-tailed, unpaired Student's *t*-tests with asterisks indicating significant differences (\*\**P* < 0.01; \*\*\*\**P* < 0.0001). Images in **b**, **d**, and **e** are representative of 4, 52, and 6 independent biological samples. Scale bars in **b**, **d**, and **e** are 10 μm, and in **g** the scale bars are 100 μm.

at a rate of 0.39 h$^{-1}$ (Fig. 2b). Salmonella, with the coupled *PsseJ-LysE* construct and that constitutively expresses GFP (*white arrow*), lysed after invasion into cancer cells and discharged GFP (*black arrow*) into the cytoplasm (Fig. 2c). Most (68%) *PsseJ-LysE* Salmonella lysed after invasion, which was significantly greater than *PsseJ-GFP* controls that did not lyse ($P < 0.0001$, Fig. 2d). After invasion into cancer cells, the engineered bacteria lysed over the course of 10 h (Fig. 2e). During this period, the cumulative fraction of cancer cells with intact bacteria decreased exponentially at a rate of 0.33 h$^{-1}$, which is equivalent to a half-life of 2.1 h (Fig. 2e). The basal expression of Lysin E by the *PsseJ-LysE* circuit did not affect bacterial health and intracellular induction activated lysis at near to its maximum rate (Fig. 2f). Based on immunoblot of lysed Salmonella, each bacterium contained, on average, 163,000 molecules of GFP, indicating how much protein can be delivered (Fig. 2g).

We defined the platform strain that uses the *PsseJ-LysE* construct to deliver proteins as intracellular delivering (ID) Salmonella. In tumor-bearing mice, ID Salmonella invaded cells and delivered GFP that filled the entire cytoplasm of cells (*black arrows*, Fig. 2h). The ID strain natively expresses *flhDC* and naturally invades into cells. Using selective permeabilization to isolate delivered protein and not include unreleased bacterial protein, it was determined that no GFP was delivered to the livers or spleens of any mice (Fig. 2i). Comparison to a GFP standard showed that ID Salmonella delivered $60 \pm 12$ μg GFP/g tumor, which is equivalent to $1.5 \times 10^8$ bacteria per gram of tumor. To demonstrate specific targeting of an intracellular protein, cancer cells were administered ID Salmonella that delivered an anti-actin nanobody (Fig. 2j). After invasion and lysis, the delivered nanobody (NB) was bound to cellular actin.

**Protein release from ID Salmonella and SCVs.** To determine the mechanisms of protein release in cells a specialized technique was

developed to identify lysed Salmonella and delivered GFP (Fig. 3a, b). Treatment with a mild detergent selectively permeabilized the membranes of mammalian cells while leaving bacterial membranes intact. In cells administered ID Salmonella with *PsseJ-LysE* (Fig. 3a, top), bacteria lysed (*black arrows, faint red*), and delivered GFP (*bright green*). Intact bacteria (*white arrows, bright red, not green*) could be easily distinguished from lysed bacteria. The faint red structures are membranes of lysed Salmonella. In cells administered nonlysing control Salmonella, only intracellular bacteria, and not the GFP in the bacteria, was visible (Fig. 3a, bottom). The amount of GFP detected in cultures administered ID Salmonella was fifty times greater than control cultures ($P < 0.0001$, Fig. 3b), showing the selectivity of the method.

When ID Salmonella delivered protein to cells, it was first released into SCVs and then dispersed into the cytoplasm (Fig. 3c–h). Immediately after invasion, most Salmonella (*white arrow*) resided within LAMP1-stained SCVs (*yellow arrow*, Fig. 3c). After several hours, lysis of ID Salmonella released GFP into SCVs (*yellow arrow*), where it was retained by the vacuole membrane and was accessible to staining (*black arrow, green dots*, Fig. 3d). At 6 h after invasion, most GFP was contained in SCVs (Fig. 3e, left), but by 24 h, the GFP had escaped the SCVs and was dispersed throughout the cytoplasm (Fig. 3e, right). This migration of GFP from SCV to cytoplasm occurred in most cells ($P < 0.0001$, Fig. 3f). From the lysis of a group of Salmonella (Fig. 3g and Supplementary Movie 1), the effective diffusivity of GFP through the cytoplasm was calculated to be 0.15 μm$^2$/min (Fig. 3h).

As designed, lysis occurred because Salmonella reside within SCVs (Fig. 3i–k). This dependence on SCVs can be seen in the small population of Salmonella that escape into the cytoplasm and are not surrounded by a SCV membrane (*white arrow*, Fig. 3i). ID Salmonella in LAMP1-stained SCVs (*yellow arrow*) released GFP (*black arrow*, Fig. 3i). In comparison, ID Salmonella in the cytoplasm (*light blue*) did not release GFP (Fig. 3i). Across multiple cells, more than 95% of lysed Salmonella were inside

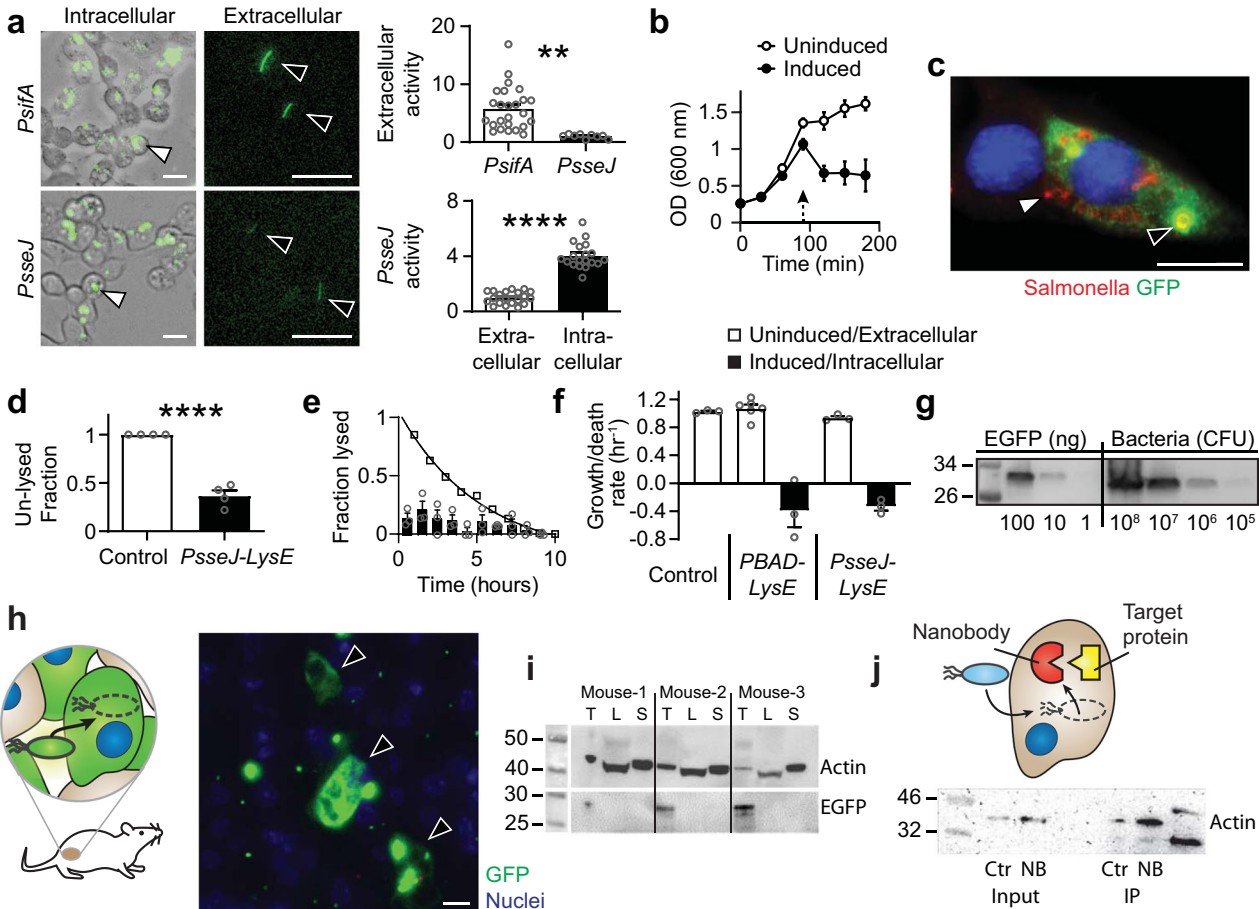

**Fig. 2 Design of ID Salmonella to release protein into cells. a** Salmonella with the *PsifA-GFP* or *PsseJ-GFP* reporter constructs both expressed GFP after invasion (*white arrows*). Extracellular expression (*black arrows*) from *PsseJ-GFP* was less than *PsifA-GFP* (P = 0.005; n = 27 for *PsifA* and n = 9 for *PsseJ*). The intracellular activity of the *PsseJ* promoter was four times greater than extracellular activity (P < 0.0001; n = 20). **b** Induction of *PBAD-LysE* at 96 h (*arrow*) induced bacteria lysis (n = 3). **c** When administered to MCF7 cancer cells, Salmonella (*red, white arrow*) with *PsseJ-LysE* and *Plac-GFP* delivered GFP (*green, black arrow*) into the cellular cytoplasm. Only released, and not intrabacterial GFP was stained. **d** Most intracellular Salmonella with *PsseJ-LysE* lysed, which was significantly greater than *PsseJ-GFP* controls (P < 0.0001; n = 4). **e** Lysis of intracellular *PsseJ-LysE* Salmonella occurred for 10 h after invasion (n = 3). The fraction of MCF7 cancer cells in which Salmonella lysed (disappeared) per hour (*bars*) was relatively constant. The cumulative fraction of cells with intact bacteria (*open circles*) decreased exponentially. **f** In liquid culture, *PBAD-lysE* and *PsseJ-LysE* Salmonella grew at similar rates as nontransformed controls (*white bars*). When intracellular, *PsseJ-LysE* Salmonella, lysed at a similar rate as induced *PBAD-lysE* Salmonella in culture (*black bars*; n = 3 for all conditions except n = 6 for uninduced *PBAD-lysE*). **g** The EGFP content of *PsseJ-LysE* Salmonella was determined by immunoblot. Lysed bacteria at $10^6$, $10^7$, $10^8$, and $10^9$ colony forming units (CFU) per well (*four right lanes*) were compared to EGFP standards at three concentrations: 1, 10, and 100 ng/well (*three left lanes*). **h** Ninety-six hours after intratumoral injection of $2 \times 10^6$ bacteria/mouse to BALB/c mice with subcutaneous 4T1 tumors (n = 3), ID Salmonella (which utilizes *PsseJ-LysE*) delivered GFP into cancer cells (*arrows*). **i** In the same mice, delivered GFP was present in extracts from tumors (T), but not livers (L) or spleens (S). Actin was used as a loading control. **j** Anti-actin nanobody (NB) and GFP (Ctr) was delivered into 4T1 cancer cells with ID Salmonella. Beta-actin was immuno-precipitated with delivered nanobody and was enriched 2.5 times compared to controls. Data are shown as means ± SEM. Statistical comparisons in **a** and **d** are two-tailed, unpaired Student's *t*-tests with asterisks indicating significance (**P < 0.01; ****P < 0.0001). Images in **a**, *left* and **c** are representative of 20 and 10 independent biological samples. The immunoblot in **j** is from a single experiment and the immunoblots in **g** and **i** are each representative of two independent experiments with similar results. Scale bars in **a**, **c**, and **h** are 10 μm.

SCVs (P < 0.0001; Fig. 3j). The dependence on SCVs was further quantified with ID Salmonella with Δ*sifA* and Δ*sseJ* deletions, which are predominantly cytoplasmic[50] and vacuolar[51], respectively (Fig. 3k). After invasion into cancer cells, Δ*sifA* ID Salmonella did not lyse (red, Fig. 3k, upper left), whereas almost all Δ*sseJ* ID Salmonella lysed (green, upper right). Without either deletion, some ID Salmonella did not lyse (red, Fig. 3k, lower left), although most localized to SCVs, lysed and delivered protein (*green*). Both ID and Δ*sseJ* ID Salmonella lysed significantly more than Δ*sifA* ID Salmonella (P < 0.0001, Fig. 3k, lower right). This dependence indicates that the *PsseJ* promoter in the *PsseJ-LysE* construct only activates in SCVs and not in the cytoplasm.

**Control of invasion to deliver proteins into cells within tumors.** A strain of ID Salmonella was constructed to enable control of *flhDC* and cell invasion. The Δ*flhD* Salmonella strain was transformed with both the *PsseJ-LysE* and *PBAD-flhDC* gene circuits (Supplementary Table 1) and named IDf+ Salmonella (Fig. 4a). Both the *PsseJ-LysE* circuit and expression of *flhDC* were required for intracellular delivery (Fig. 4b, lower right). IDf+ Salmonella with uninduced *PBAD-flhDC* neither invaded nor released protein (Fig. 4b, upper and lower left). Induction of *PBAD-flhDC* without *PsseJ-LysE* lead to invasion but not delivery (Fig. 4b, upper right). In cells stained for released GFP, only those administered IDf+ Salmonella with both induced *PBAD-flhDC*

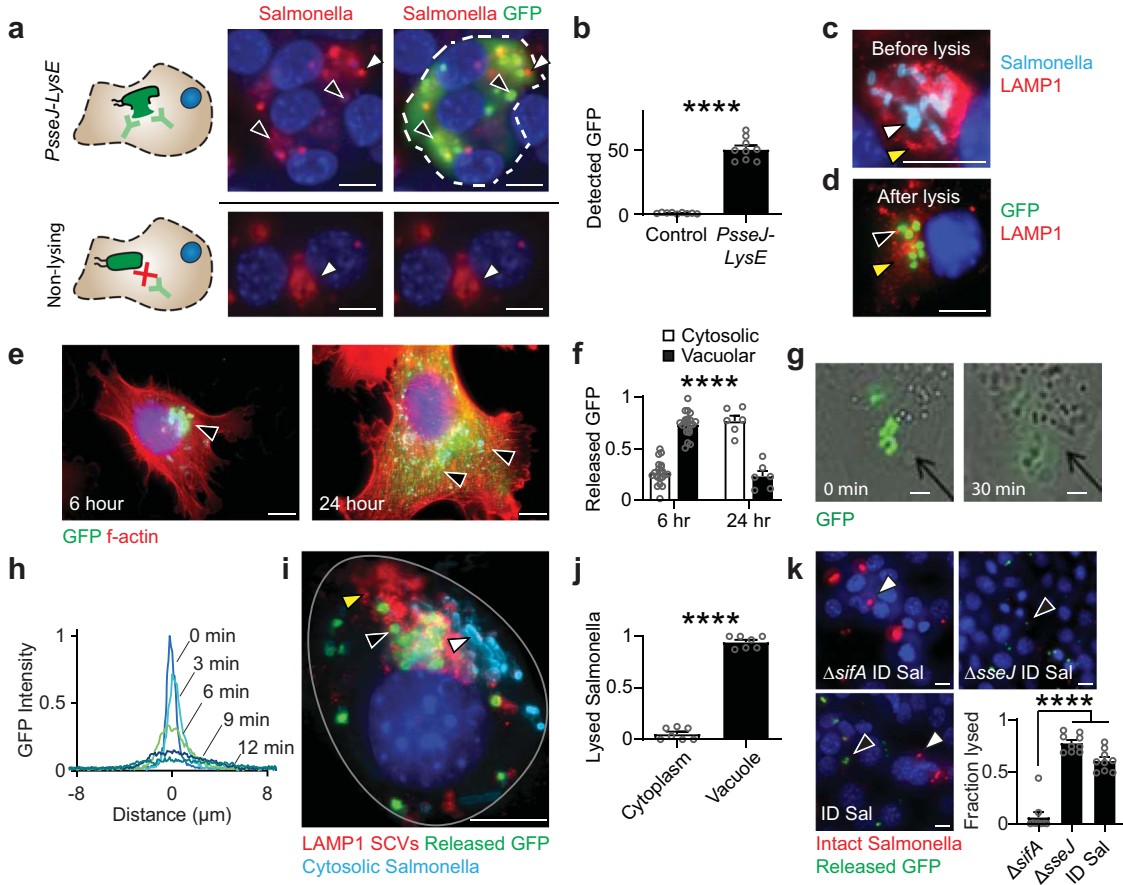

**Fig. 3 Release of delivered proteins from SCVs. a** Selective permeabilization of cancer cell membranes enabled detection of released GFP. After administration to cells at an MOI of 10, ID Salmonella (*top*) released GFP (*green, black arrows*). Intact bacteria (*red, white arrows*) are easily discernable from the membranes of lysed bacteria (*faint red, black arrows*). In cells administered nonlysing Salmonella that produced GFP (*bottom*), only intracellular bacteria (*red, white arrows*) and no GFP was detected. **b** The amount of GFP detected in cultures administered ID Salmonella was fifty times greater than nonlysing controls (*P* < 0.0001; *n* = 9). **c** After invasion and before lysis, Salmonella (*light blue, white arrow*) are in LAMP1-stained SCVs (*red, yellow arrows*). **d** After lysis, GFP (*green, black arrow*) is retained within the membranes of SCVs (*red, yellow arrow*). **e** In phalloidin-stained cancer cells (*red*), released GFP (*green, black arrows*) moved from SCVs near the nucleus (*blue*) to throughout the cytoplasm. **f** From 6 to 24 h after invasion, the percentage of released GFP in the cytosol increased from 25 to 75% (*P* < 0.0001; *n* = 21 at 6 h and *n* = 6 at 24 h). **g** The lysis of individual ID Salmonella released GFP that diffused through the cytosol of a cancer cell (see Supplementary Movie 1). **h** Temporal profiles of GFP intensity, centered on the lysed bacteria. **i** Salmonella in LAMP1-stained SCVs (*red, yellow arrow*) lysed and released GFP (*green, black arrow*). Cytosolic bacteria (*light blue, white arrow*) did not lyse. **j** Most lysed bacteria, identified by colocalized staining for Salmonella and released GFP, were located within LAMP1-stained SCVs (*P* < 0.0001; *n* = 7). **k** Predominantly cytoplasmic Δ*sifA* ID Salmonella remained intact (*red, white arrows*) and lysed less (*green, black arrows*) than predominantly vacuolar Δ*sseJ* ID Salmonella and ID Salmonella (*P* < 0.0001; *n* = 9). Data are shown as means ± SEM. Statistical comparisons in **b**, **f**, and **j** are two-tailed, unpaired Student's *t*-tests. The statistical comparisons in **k** were to a single control performed with ANOVA followed by Dunnett's method. Two outliers were removed from Δ*sifA* using the ROUT method with a Q of 1%. Asterisks indicate significance (****P* < 0.0001). Images in **a**, **c**, **d**, **i**, and **k** are representative of 9, 25, 7, 18, and 9 independent biological samples. Images in **e** are representative of 21 and 6 independent biological samples at 6 and 24 h, respectively. The images in **g** are frames from Supplementary Movie 1. Scale bars in **a**, **c**–**e**, **i**, and **k** are 10 μm, and in **g** the scale bars are 1 μm.

and *PsseJ-LysE* delivered GFP (Fig. 4c). This strain delivered 548 times more GFP than controls (*P* < 0.0001; Fig. 4d).

In tumors, delivery of protein could be controlled with *flhDC* (Fig. 4e, f). After administration of Salmonella that did not express *flhDC* to tumor-bearing mice, little GFP was delivered (Fig. 4e, left). Re-expressing *flhDC* after initial colonization increased the number of cells that received GFP (Fig. 4e, right). In the transition zone of tumors with induced *flhDC*, 21% of cells received GFP (*P* < 0.001), which was five times greater than controls (*P* < 0.001, Fig. 4f).

**Safety of ID Salmonella.** Delivery of proteins with ID Salmonella is self-limiting and nontoxic (Fig. 5). Mice with orthotopic mammary tumors were intravenously injected with luciferase-expressing ID Salmonella (Fig. 5a). The bacterial density in the tumors initially increased and then decreased, reaching a peak at 72 h (Fig. 5a, b). The density then dropped 95% by day 14 (*P* < 0.05, Fig. 5b). This decline in density eliminates the bacteria from the mouse after it has delivered the protein payload. In healthy, tumor-free mice, ID Salmonella did not accumulate in the lungs, hearts, kidneys, or brains (Fig. 5c and S2). After intravenous injection, Salmonella were present in several organs at 6 h, but were cleared by 7 days (Supplementary Fig. 2). At 14 days, the bacterial density in livers and spleens was 3750 and 14,100 times less than in tumors, respectively (*P* < 0.05, Fig. 5c). Comprehensive hematology showed that injection of ID Salmonella did not affect the number of immune cells in the blood compared to saline controls (Fig. 5d and S1a). Chemistry

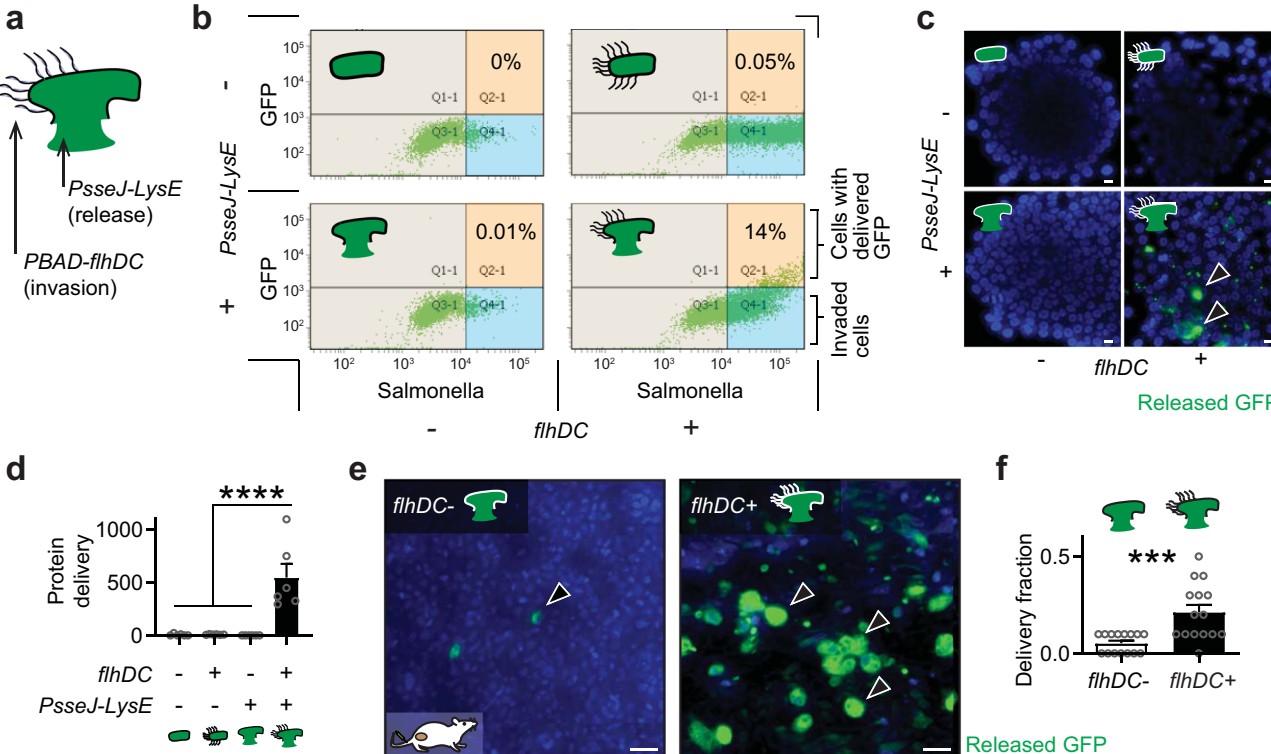

**Fig. 4 PsseJ-LysE and flhDC are necessary for delivery to tumors. a** To control cell invasion and protein release, IDf+ Salmonella were created by transforming ΔflhD Salmonella with PsseJ-LysE, PBAD-flhDC, and Plac-GFP. **b** IDf+ Salmonella were administered to 4T1 cancer cells and PBAD-flhDC was induced with 20 mM arabinose. Controls either lacked PsseJ-LysE, were uninduced, or both. Invasion was detected with anti-Salmonella antibodies and delivery was detected by the presence of released GFP. **c** GFP (green, arrows) was only delivered from Salmonella with activated PBAD-flhDC and transformed with PsseJ-LysE. **d** Induced IDf+ Salmonella delivered significantly more GFP than all controls (P < 0.0001; n = 6). **e** IDf+ Salmonella (2 × 10^8 CFU/mouse) were intravenously injected into BALB/c mice with subcutaneous 4T1 tumors via the tail vein (n = 3). In flhDC+ mice, 100 μg of arabinose was injected IP at 48 and 72 h. At 96 h, delivered GFP (arrows) was measured in excised tumors with immunohistochemistry. **f** In the transition zone of the tumors in **e**, induction of flhDC increased the fraction of cells with delivered GFP (P = 0.0004; n = 15). Data are shown as means ± SEM. The statistical comparisons in **d** were to a single condition performed with ANOVA followed by Dunnett's method. The statistical comparison in **f** is a two-tailed, unpaired Student's t-test. Asterisks indicate significance (***P < 0.001; ****P < 0.0001). The results in **b** are from a single experiment and the images in **c** are representative of 6 independent biological samples. Scale bars in **c** and **e** are 10 μm.

profiling showed that ID Salmonella do not cause liver damage (Fig. 5e and S1b).

**Intracellular delivery of protein drugs.** To demonstrate its therapeutic capabilities, ID Salmonella was engineered to produce two protein drugs, NIPP1-CD and CT Casp-3. When delivered to cancer cells in an invasion assay, NIPP1-CD caused more death than control ID Salmonella without NIPP1-CD (P < 0.01; Fig. 6a). In microfluidic tumors devices, ID Salmonella that deliver NIPP1-CD (*NIPP1-CD* Salmonella) caused cell death (red), which increased with time (P < 0.05, Fig. 6b). Across whole tissues, NIPP1-CD caused four times more death than controls (P < 0.05; Fig. 6c). In mice with 4T1 tumors, ID Salmonella (black arrow, green) delivered NIPP1-CD (white arrow, red) across the tumor tissue (Fig. 6d). From images of mice that received *NIPP1-CD* Salmonella, ~23 ± 5% of cells contained delivered NIPP1-CD. Despite effective delivery, *NIPP1-CD* Salmonella did not affect tumor growth (Supplementary Fig. 3).

ID Salmonella expressing CT Casp-3 (*CT Casp-3* Salmonella) caused Hepa 1–6 cells to die in culture (Fig. 7a) eight times more than controls (P < 0.0001, Fig. 7b). The induced death was dependent on cell invasion and protein delivery. Cells died (white arrow, red, lower right) 10 h after Salmonella invasion and CT Casp-3 delivery (green, lower left). Cells that were not invaded and did not receive CT Casp-3 (yellow arrow) did not die.

Similarly, cells that were invaded by control ID Salmonella and only delivered GFP (black arrow, upper left) did not die. In microfluidic tumors devices (Fig. 7c), CT Casp-3 caused twice as much cell death as bacterial controls (P < 0.01, Fig. 7d). Although control bacteria kill some cells in these devices (and as seen previously[52]), the difference in cell death is due to the delivery of CT Casp-3.

Delivery of CT Casp-3 was effective against both liver cancer and triple-negative breast cancer in mice (Fig. 7e–j). After 14 days of treatment, delivery to BALB/c mice reduced the volume of 4T1 mammary tumors two times more than controls (P < 0.05, Fig. 7e). Bacterial delivery of CT Casp-3 was also nontoxic. When injected into tumor-free mice, *CT Casp-3* Salmonella was cleared from most organs in 7 days (Supplementary Fig. 2) and did not induce any toxicity compared to either saline or empty ID Salmonella (Supplementary Fig. 4). In BALB/c mice, treatment with CT Casp-3 blocked the growth of established 4T1 metastases in the lung (P < 0.05, Fig. 7f, g). In comparison, after treatment with 10 mg/kg paclitaxel, the standard-of-care, metastatic volume increased 85 times. No metastases grew in any mice treated with CT Casp-3 (Supplementary Fig. 5).

Administration of *CT Casp-3* Salmonella significantly reduced the growth of two liver cancer models: BNL-MEA and Hepa 1–6. Intravenous injection of *CT Casp-3* Salmonella reduced the growth of BNL-MEA tumors by 47 and 57% compared to saline

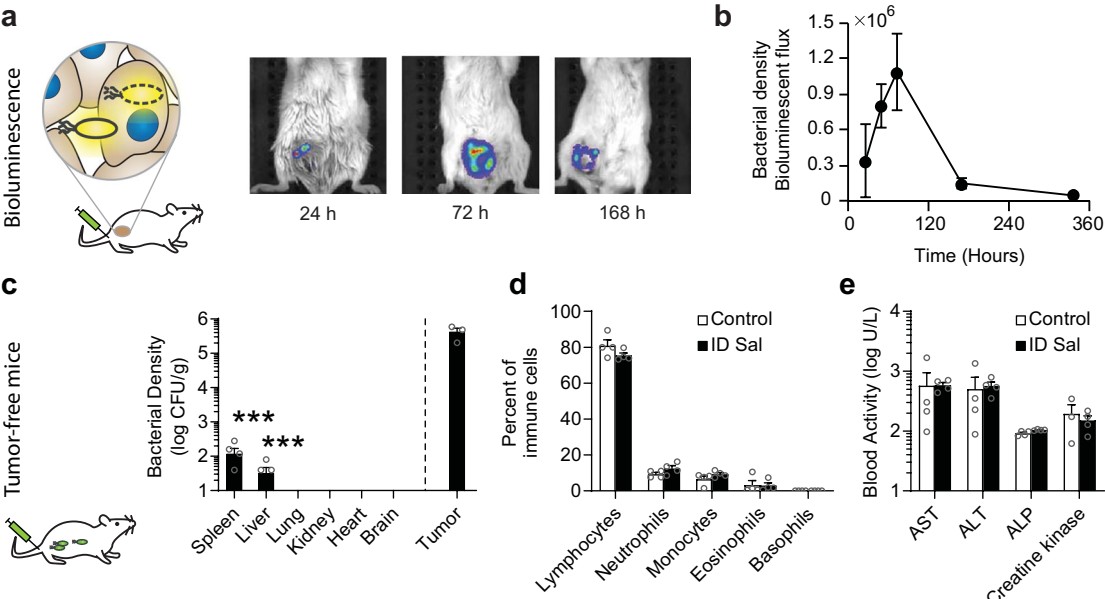

**Fig. 5 Safety, biodistribution, and clearance of ID Salmonella. a** Firefly luciferase-expressing ID Salmonella ($2 \times 10^7$ CFU/mouse) were intravenously injected into BALB/c mice with 4T1 tumors in the mammary fat pad ($n = 4$). Prior to imaging, mice were injected IP with 100 µl of 30 mg/ml D-luciferin. **b** The bacterial density in the tumors increased for 72 h and then decreased. **c** Biodistribution of bacteria in tumor-free BALB/c mice, 14 days after intravenous injection with $1 \times 10^7$ ID Salmonella ($n = 5$). Densities were below detection in the lungs, kidneys, hearts, and brains. Measurements of zero bacteria in spleen (1 of 5) and liver (2 of 5) were not displayed. In spleens and livers, bacterial densities were more than 3000 times lower than in tumors (from the separate experiment in panel **b** using the same organ mincing technique ($P = 0.0001$, $n = 3$ mice, one mouse died prior to density measurement). **d** Comprehensive hematology of blood drawn from tumor-free BALB/c mice, 14 days after intravenous injection with $1 \times 10^7$ ID Salmonella or saline ($n = 4$). No changes were observed in the number of any immune cells in the blood. **e** Chemistry profiling of the blood from the mice in **d**. There was no indication of liver damage, despite some liver colonization (**c**). Markers of liver damage are ALP alkaline phosphatase, ALT alanine transaminase, and AST aspartate transaminase. Data are shown as means ± SEM. Statistical comparisons in **c** were to a single condition performed with ANOVA followed by Dunnett's method and asterisks indicate significance (***$P < 0.001$).

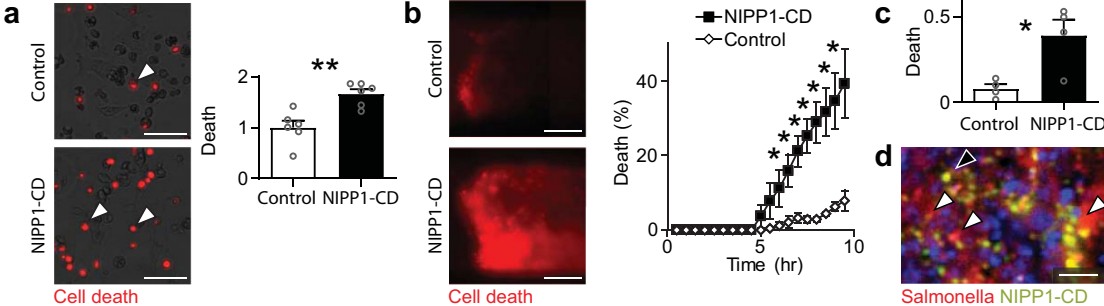

**Fig. 6 Delivery of NIPP1-CD. a** ID Salmonella delivery of NIPP1-CD caused more death (*red, white arrows*) in Hepa 1-6 cells compared to controls ($P = 0.0021$, $n = 6$). **b** In microfluidic tumor masses, delivery of NIPP1-CD caused more cell death (*red*) than bacterial controls. The percentage of dead cells increased with time as Salmonella invaded into cells and delivered protein ($P = 0.042, 0.017, 0.014, 0.017, 0.024, 0.030$, and $0.039$ at 6.5, 7, 7.5, 8, 8.5, 9, and 9.5 h, respectively, $n = 4$). **c** For multiple ($n = 4$) tumor masses, as shown in **b**, NIPP1-CD significantly increased cell death ($P = 0.0177$). **d** *NIPP1-CD* ID Salmonella was intravenously administered to BALB/c mice with subcutaneous 4T1 tumors. After 31 days, Salmonella (*black arrow, green*) and delivered NIPP1-CD (*white arrows, red*) were dispersed throughout the tissue. Data are shown as means ± SEM. Statistical comparisons in **a–c** are two-tailed, unpaired Student's t-tests with asterisks indicating significance (*$P < 0.05$; **$P < 0.01$). Images in **a, b**, and **d** are representative of 6, 4, and 3 independent biological samples, respectively. Scale bars in **a** and **b** are 100 µm, and in **d** the scale bar is 10 µm.

and Sorafenib ($P < 0.05$, Fig. 7h), which is the standard-of-care for liver cancer. Similarly, treatment with *CT Casp-3* Salmonella reduced the volume of liver Hepa 1–6 tumors in C57L/J mice ($P < 0.001$; Fig. 7i) and reduced tumor growth rate 28 times ($P < 0.05$; Supplementary Fig. 6). This reduction in growth is equivalent to an increase in doubling time from 5 to 148 days. Tumor volume reduced in two mice for over 50 days, and survival increased significantly compared to bacterial controls ($P < 0.05$, Fig. 7j). In mice that survived over 50 days, no toxic effects were observed from long-term bacterial therapy. Treatment with CT

Casp-3 completely eliminated the tumor from one mouse, which remained disease free for over 300 days.

## Discussion

We have created an autonomous, intracellular Salmonella vehicle that efficiently delivers active proteins into cells. The ID Salmonella strain is safe, reduces the growth of tumors and metastases, and increases survival in mice. The developed bacterial strains utilize three gene circuits that enable precise control of drug

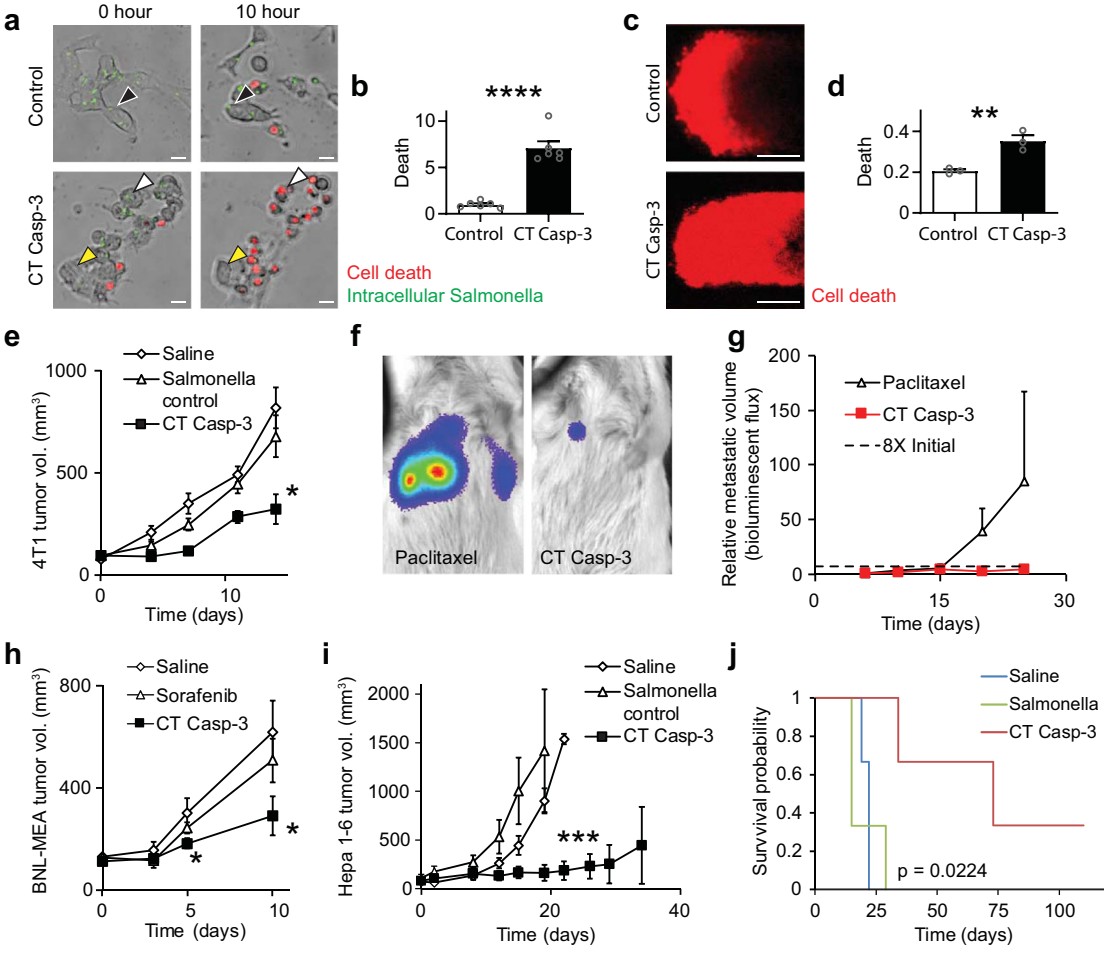

**Fig. 7 Delivery of CT Casp-3 reduced tumor growth and increased survival. a** Delivery of CT Casp-3 killed Hepa 1–6 cells (*white arrows, red*). Cells invaded with control Salmonella (*green, black arrows*) or not invaded (*yellow arrows*) did not die. **b** CT Casp-3 caused significantly more death than controls ($P < 0.0001$, $n = 6$). **c** In microfluidic tumor masses, CT Casp-3 caused more cell death (*red*) than bacterial controls. **d** The fraction of dead cells was significantly greater in masses administered CT Casp-3 compared to controls ($P = 0.0072$, $n = 3$). **e** In 4T1 subcutaneous mammary tumors, intratumoral delivery of $4 \times 10^7$ CFU of *CT Casp-3* Salmonella decreased growth compared to bacterial controls, which delivered GFP ($P = 0.0199$, $n = 6$ mice per condition). **f** Lung metastases were established by injection of $5 \times 10^4$ luciferase-expressing 4T1 cells into the tail vein of BALB/c mice. Mice were treated with either $1 \times 10^7$ CFU of *CT Casp-3* Salmonella or 10 mg/kg paclitaxel ($n = 6$), and imaged after IP injection of 100 µl of 30 mg/ml D-luciferin. Metastases grew exponentially in some paclitaxel-treated mice, but not in any treated with *CT Casp-3* Salmonella. **g** On average, metastases grew over 85 times after treatment with paclitaxel. Treatment with *CT Casp-3* Salmonella prevented growth more than eight times the initial volume (*dotted line*; $P = 0.027$). Metastatic volume is based on bioluminescence imaging and is reported relative to initial volume (see Supplementary Fig. 5 for individual mice). **h** In subcutaneous BNL-MEA liver tumors, intravenous delivery of $1 \times 10^7$ CFU of *CT Casp-3* Salmonella decreased growth compared to 10 mg/kg Sorafenib ($P = 0.045$) or saline ($P = 0.031$; $n = 5$ for saline and CT Casp-3 and $n = 4$ for Sorafenib). **i** In C57L/J mice with subcutaneous Hepa 1–6 liver tumors, intratumoral injection of $4 \times 10^7$ CFU of *CT Casp-3* Salmonella reduced volume to 11% of bacterial controls (ID Salmonella) by 19 days and 12% of saline controls by 22 days after injection ($P = 0.00083$, $n = 3$ mice per condition). **j** Delivery of CT Casp-3 significantly increased survival ($P = 0.0224$) and cured one mouse. Data are shown as means ± SEM. Statistical comparisons are one-sided (**h**) and two-sided (**b**, **d**, **e**, **g**, **i**) unpaired Student's *t*-tests, with asterisks indicating significance (*$P < 0.05$; **$P < 0.01$; ***$P < 0.001$; ****$P < 0.0001$). The statistical comparisons in **j** are log-rank tests with Bonferroni correction. Images in **a** are representative of six independent biological samples per condition. Scale bars in **a** and **c** are 20 and 100 µm, respectively.

production, cell invasion, and protein release. These processes were derived from natural capabilities of Salmonella. The *PsseJ-LysE* circuit was designed to trigger protein release independent of external intervention (Fig. 2). This self-timing system triggers delivery at the most opportune time for each bacterium and ensures that proteins are deposited inside cells and not in the extracellular environment. As a platform, ID Salmonella are capable of delivering multiple types of proteins including nanobodies and cytotoxic proteins (NIPP1-CD and CT Casp-3).

The *PsseJ-LysE* circuit makes ID Salmonella self-limiting and safer than nonlysing Salmonella. After injection, the density of ID Salmonella in tumors reaches a maximum at 72 h (Fig. 5b). After this peak, the bacteria are cleared. In comparison, our group and

others have shown that nonlysing Salmonella grow exponentially in tumors[16,53]. This peak in bacterial density is most likely caused by the timing of invasion and lysis. During the first 72 h after injection, ID Salmonella accumulate in tumor tissue, invade into cancer cells, and form SCVs. After 72 h, the Salmonella in the SCVs lyse, which releases their contents into host cells and reduces the bacterial density. This intracellular lysis would limit unintended exposure to protein therapies and reduce the possibility of unwanted infections.

Salmonella were used to create this delivery platform because of their unique physiology. Two essential qualities of a delivery species are invasiveness and ability to sense the intracellular environment. Numerous bacterial species invade into cells[54,55],

but the promoters of SPI1 and SPI2 are unique to Salmonella. The SPI2 promoters provide a tool to activate gene expression once the bacteria are in SCVs, which are explicitly intracellular. At first it would appear that residence in SCVs would be detrimental to protein delivery. Indeed, when ID Salmonella first lyse, the payload protein is released into SCVs (Fig. 3d). However, with time, the protein disperses into the cytoplasm (Fig. 3e–h), where it can interact with its targets (Fig. 4g). This dispersion is most likely mediated by the breakdown of the vacuole membrane once the bacteria are no longer viable and injecting effector proteins to maintain membrane integrity[56,57]. In this way, accumulation in SCVs provides a tool to precisely control the timing of protein release.

Control of cell invasion is another tool that could increase the safety and efficacy of protein delivery. We discovered that the regulator protein complex FlhDC is necessary for cell invasion (Fig. 1f–i) and we developed a gene circuit (PBAD-flhDC) to controls its expression (Fig. 4). In culture and tumors, activation of flhDC increased protein delivery (Fig. 4c, h, i) by forcing all bacteria to produce flagella, invade cells and lyse. Modulating flhDC also has the potential to control the location of delivery. For example, expression could be induced after bacteria have accumulated in tumors and cleared from most organs. Activating flhDC would trigger bacteria to invade nearby cancer cells and deliver their protein payload. This timing would focus delivery to tumors and prevent release of therapeutic molecules into healthy tissue.

Two essential qualities of ID Salmonella enable the use of protein drugs that are currently not feasible. Constitutively active proteins would be too toxic if delivered traditionally and limited access to intracellular compartments would reduce efficacy. Active transport across the cell membrane and specific accumulation in tumors solves both of these problems. The example proteins, NIPP1-CD and CT Casp-3, have exclusively intracellular targets and would be ineffectual without intracellular delivery. Both of these proteins were modified from their natural counterparts to be constitutively active and would be toxic if delivered into all cells systemically. Importantly, CT Casp-3 activates apoptotic pathways that are circumvented by most cancer cells[48]. Molecules that have ubiquitous targets would be effective in most cancer types and would be independent of a cancer's mutational profile. Specific accumulation in tumors ensures the safety of the therapy and prevents adverse events (Figs. 5 and S4).

The differential response of tumors to delivery of CT Casp-3 and NIPP1-CD suggests that efficacy is dependent on many factors, and not just cellular cytotoxicity. In culture, delivery of both molecules killed cancer cells (Figs. 6a–c and 7a–d); however, NIPP1-CD was not effective in mice (Supplementary Fig. 3). This suggests that the response to a bacterial delivered protein is dependent on the cellular environment, other tumor-associate cells, and the immune response. These dependences may also be uniquely affected by the presence of tumor-colonized bacteria.

The use of ID Salmonella to deliver CT Casp-3 could address the need for an effective treatment for unresectable hepatocellular carcinoma (HCC). No curative treatment currently exists for the 840,000 patients who are diagnosed with HCC annually[58,59]. Current therapies have toxic side effects and only modestly increase survival[59–61]. Treatment with CT Casp-3 Salmonella could be safer (Figs. 5, S3, and S4) and more effective (Fig. 7). By seven days after injection, CT Casp-3 Salmonella were cleared from most organs (Supplementary Fig. 2b). The presence of bacteria in organs shortly after injection (6 h, Supplementary Fig. 2a) has been commonly observed, and these organisms are predominantly in the blood and not interacting with tissues[17]. Although trace amounts of bacteria were present in the spleen at 7 days, autonomous lysis caused this amount to be about

1000 times lower than previous measurements[17,53]. In addition, no delivered proteins were detected in healthy tissue (Fig. 4f) and no toxicities were observed from the bacteria or the CT Casp-3 (Supplementary Fig. 4).

Delivery with ID Salmonella will enable targeting of inaccessible cancer pathways and will accelerate the generation of cancer therapies. These therapies can be created by coding the genes for specific protein drugs into Salmonella expression cassettes. Nanobodies (Fig. 4g) can be designed that specifically inhibit pathways necessary for cancer survival and progression. The efficacy in a metastatic breast cancer model (Fig. 7f, g) suggests that ID Salmonella finds and halts metastases and could be used to treat advanced disease. Using bacteria to deliver proteins into cells will expand the number of accessible pathways, open up many targets across the soluble proteome for treatment, and increase the efficacy and safety of cancer treatment.

## Methods

**Bacterial strains and plasmid construction.** Fifteen strains of *Salmonella enterica* serovar Typhimurium were used throughout the experiments (Supplementary Table 1). The parental control strain (*Par*) is derived from an attenuated strain of Salmonella (VNP20009) and has four deletions, ΔmsbB, ΔpurI, Δxyl, and Δasd. All plasmids contained a ColE1 origin and either chloramphenicol or ampicillin resistance (Supplementary Table 2). Three additional genomic knockouts (ΔflhD, ΔsifA, and ΔsseJ) were created using a modified lambda red recombination protocol[62] and primers with specific homology regions (Supplementary Table 3). Plasmids (Supplementary Table 2) were inserted into these base strains to generate strains that produce GFP (Accession KP294373) after cell invasion, re-express flhDC (Accession CP001363 [https://www.ncbi.nlm.nih.gov/geo/query/acc.cgi?acc=GPL14855]), report activation of PsifA (Accession CP001363 [https://www.ncbi.nlm.nih.gov/geo/query/acc.cgi?acc=GPL14855]), and produce Lysin E (Accession AF176034) after activation of PsseJ (Accession CP001363 [https://www.ncbi.nlm.nih.gov/geo/query/acc.cgi?acc=GPL14855]). The ID Salmonella strain was transformed to express a nanobody against β-actin (*Chromotek*), NIPP1-CD (Accession NM_174582.2), and CT Casp-3 (Accession AY219866). Details of gene deletions and plasmid construction are described below and the primers used are in Supplementary Tables 3 and 4.

All bacterial cultures (both Salmonella and DH5α) were grown in LB (10 g/L sodium chloride, 10 g/L tryptone and 5 g/L yeast extract). Resistant strains of bacteria were grown in the presence of carbenicillin (100 μg/ml), chloramphenicol (33 μg/ml), kanamycin (50 μg/ml), and/or 100 μg/ml of diamino-pimelic acid (DAP). All assembled DNA constructs were transformed into chemically competent DH5α E. Coli (*New England Biolabs*, Ipswich, MA) before electroporation into Salmonella. Electroporation was performed in 1 mm cuvettes at 1800 V and 25 μF with a time constant of 5 msec. All cloning reagents, buffer reagents, and primers were from *New England Biolabs, Fisher Scientific* (Hampton, NH), and *Invitrogen*, (Carlsbad, CA), respectively, unless otherwise noted.

**Cell culture and animal models.** Five cancer cell lines were used: 4T1 murine breast carcinoma cells; Hepa 1–6 and BNL-MEA (BNL 1ME A.7 R.1) murine hepatocellular carcinoma cells; MCF7 human breast carcinoma cells and LS174T human colorectal carcinoma cells (*ATCC*, Manassas, VA). The mouse cell lines were authenticated with CO1 barcoding and the human cell lines were authenticated with short tandem repeat profiling. All cancer cells were grown and maintained in Dulbecco's Minimal Eagle Medium (DMEM) containing 3.7 g/L sodium bicarbonate and 10% fetal bovine serum. For microscopy studies, cells were incubated in DMEM with 20 mM HEPES buffering agent and 10% FBS. To generate tumor spheroids, single-cell suspensions of LS174T cells were transferred to PMMA-coated cell culture flasks [2 g/L Poly(2-hydroxy ethyl) methylacrylate (PMMA) in 100% ethanol, dried before use].

Multiple tumor models in mice (*Mus musculus*) were used. Both male and female mice, aged 4–7 weeks, were used. Delivery mechanisms and treatment efficacy were determined using subcutaneous syngeneic tumors formed with (1) 4T1 murine breast cancer and (2) BNL-MEA liver cancer cells implanted in BALB/c mice, and (3) Hepa 1–6 murine liver cancer cells in C57L/J mice. Clearance was determined in orthotopic 4T1 tumors implanted in the mammary fat pad of BALB/c mice. Toxicity and biodistribution were determined in tumor-free BALB/c and C57L/J mice. The effect on metastases was determined in BALB/c mice intravenously injected with 4T1 cells. All animal procedures complied with relevant ethical regulations and protocols were approved by the UMass Institutional Animal Care and Use Committee (IACUC). Mice were housed under a 12 h light/dark cycle at controlled room temperature of 72 °F and a relative humidity of 60%.

**Gene deletions.** Four genetic deletions were created (Δasd, ΔflhD, ΔsifA, and ΔsseJ) using a modified lambda red recombination protocol[62]. A parental strain (*Par*) was derived from Salmonella strain VNP20009, (ΔmsbB, ΔpurI, Δxyl) by

deleting *asd*. Salmonella were transformed with pkd46 (Yale CGSC *E. Coli* stock center), centrifuged at 3000 × *g* and resuspended in ice-cold water. A PCR product was created to insert an in-frame deletion into the *asd* gene by PCR amplifying the FRT-CHLOR-FRT sequence from plasmid pkd3 (Accession AY048742.1) using primers vr266 and vr268 (Supplementary Table 3), which contain 50 basepair regions homologous to *asd*. This linear segment was transformed into Salmonella by electroporation. After recovery, colonies were screened for knockouts by colony PCR of the junction sites of the inserted PCR amplified products. Successful transformants were grown overnight at 43 °C to eliminate pkd46.

A similar process was used to delete *flhD*, *sifA*, and *sseJ*. Deletion of *flhD* prevents the formation and function of the hetero-oligomeric FlhDC complex[63]. Linear DNA segments were designed to insert in-frame deletions into the genes by amplification of the FRT-KAN-FRT sequence from plasmid pkd4 (Accession AY048743.1). Three sets of primers (vr121 and vr309 for *flhD*; vr432, and vr433 for *sseJ*; and vr434 and vr435 for *sifA*) added 50-basepair flanking regions that were homologous to the three genes (Supplementary Table 3). After electroporation and recovery, colonies were screened for knockouts by colony PCR. Successful transformants were plated on kanamycin plates (50 μg/ml) and grown overnight at 43 °C to remove pkd46.

**Plasmid construction**. Fifteen strains of Salmonella (Supplementary Table 1) were created by transforming twelve plasmids (Supplementary Table 2) into the parental strain (*Par*) and the gene knockout strains described above (i.e., Δ*flhD*, Δ*sifA*, and Δ*sseJ*). All of the plasmids contained a ColE1 origin and either chloramphenicol or ampicillin resistance (Supplementary Table 2). The intracellular-reporting strain of Salmonella was generated by transforming the parental strain (*Par*) with a plasmid containing *PsseJ-GFP* (plasmid P1; Supplementary Table 2). The construction of this plasmid was initiated by first creating a promoterless-GFP plasmid from pLacGFP and pQS-GFP[64]. The pQS-GFP plasmid (Accession KP294373) contains chloramphenicol resistance, the ColE1 origin of replication, and the *asd* gene (Accession CP001363 [https://www.ncbi.nlm.nih.gov/geo/query/acc.cgi?acc=GPL14855]). Expression of ASD is necessary in Δ*asd* strains and creates a balanced lethal system that maintains gene expression in vivo. The *Plac-GFP* gene circuit (Accession KP294375) was amplified from plasmid pLacGFP with primers nd1 and nd2 (Supplementary Table 4). The PCR product and the plasmid were digested with Aat2 and Pci1 and ligated with T4 DNA ligase (*NEB*, catalog # M0202S). The *PsseJ* promoter (Accession CP001363 [https://www.ncbi.nlm.nih.gov/geo/query/acc.cgi?acc=GPL14855]) was amplified from the genome of SL1344 Salmonella using primers nd3 and nd4 (Supplementary Table 4). This PCR product and the backbone plasmid were ligated after digestion with XbaI and Pci1.

A strain that re-expresses *flhDC* (*flhDC* Sal, Supplementary Table 1) was created by transforming Δ*flhD* Salmonella with plasmid P2 (Supplementary Table 2). Plasmid P2 was formed from temporary plasmid P3. Plasmid P3 was formed by amplifying *flhDC* (Accession CP001363 [https://www.ncbi.nlm.nih.gov/geo/query/acc.cgi?acc=GPL14855]) from Salmonella genomic DNA using primers vr46 and vr47 (Supplementary Table 4), and ligating it into plasmid PBAD-his-mycA (*Invitrogen*; catalog # V430-01). The PCR product was digested with NcoI, XhoI and DpnI (*NEB*, catalog #s R0193S, R0146S and R0176L). The PBAD-his-myc plasmid (Accession X81838) was digested with NcoI and XhoI and treated with calf intestinal phosphatase (*NEB*, catalog # M0290) for 3 h. The PCR product was ligated into the plasmid backbone with T4 DNA ligase (*NEB*, catalog # M0202S).

The *Plac-GFP-myc* circuit was inserted into P3 by Gibson Assembly. [1] The insert (*Plac-GFP-myc*) was amplified from plasmid pLacGFP[64] using primers vr394 and vr395 (Supplementary Table 4), which added homology regions to the backbone and added the *myc* tag. [2] The backbone plasmid (P3) was amplified using primers vr385 and vr386, which added homology to the insert. [3] Both PCR products were digested with DpnI for 3 h, [4] and ligated by Gibson Assembly (HiFi master mix, *NEB*, catalog # E2621L). The gene for aspartate semialdehyde dehydrogenase (*asd*) gene was inserted by Gibson Assembly by amplifying *asd* from genomic Salmonella DNA using primers vr424 and vr425, and amplifying the plasmid backbone with primers vr426 and vr427.

A strain that re-expresses *flhDC* and produces GFP after invasion (*flhDC* reporting, Supplementary Table 1) was created by transforming Δ*flhD* Salmonella with plasmid P4 (Supplementary Table 2). The *PsseJ-GFP-myc* genetic circuit was amplified from P1 using primers vr269 and vr270, and the backbone of plasmid P3 was amplified using primers vr271 and vr272. The two PCR products were ligated by Gibson Assembly.

To generate the *PsifA* intracellular promoter-reporter strain, the *PsifA* promoter (Accession CP001363 [https://www.ncbi.nlm.nih.gov/geo/query/acc.cgi?acc=GPL14855]) was cloned from Salmonella genomic DNA using primers nd5 and nd6 and inserted into P1 using XbaI and Pci1 creating plasmid P5. The *PsifA* reporter strain was created by transforming plasmid P5 into background Salmonella by electroporation. The generation of the *PsseJ* reporter strain is described above. To investigate lysis in Salmonella, lysis gene E (*LysE*) was put under control of PBAD. LysE was cloned using primers nd7 and nd8 and inserted into pBAD/*Myc*-His A (*Invitrogen*) using NcoI and KpnI to form plasmid P6.

Intracellular delivering (ID) Salmonella were created by cloning the Lysin E gene (Accession AF176034) behind the *PsseJ* promoter. *LysE* was amplified using primers nd9 and nd10 and cloned into P1 using XbaI and Aat2. The *Plac-GFP*

circuit was added to this plasmid by cloning it from plasmid pLacGFP using primers nd11 and nd12 and inserting using SacI to create plasmid P7. This plasmid constitutively expresses *myc*-tagged GFP to identify bacteria in both live-cell and fixed-cell assays.

Two strains of ID Salmonella with deletions of the *sseJ* and *sifA* genes were created by transforming the knockout strains described above (Δ*sifA* and Δ*sseJ*) with plasmid P7. Similar to ID Salmonella, these two strains contain the *PsseJ-LysE* construct and constitutively express *myc*-tagged GFP. Note the distinction between the effector gene *sseJ*, which is necessary for vacuolar escape, and its promotor *PsseJ*, which activates in SCVs.

ID Salmonella that re-expresses *flhDC* (IDf + Sal) was created by transforming Δ*flhD* with plasmids P8. Plasmid P8 was created by amplifying the *PsseJ-LysE* gene circuit from P7 using primers vr398 and vr399, and ligating it into plasmid P2 using Gibson Assembly. The P2 backbone plasmid was amplified using primers vr396 and vr397.

A strain of ID Salmonella that constitutively expresses luciferase (ID Sal-luc; Supplementary Table 1) was created by cloning *Plac-luc* from pMA3160 (*Addgene*) using primers ch1 and ch2. The P7 plasmid backbone was amplified with primers ch3 and ch4 and the pieces were ligated by Gibson Assembly to form plasmid P9 (Supplementary Table 2).

To create ID Salmonella that express anti-b-actin nanobody (NB), PBAD-inducible nanobody was cloned in place of *flhDC* in plasmid P8. The actin nanobody (*Chromotek*, catalog # acr) was amplified using primers vr466 and vr467. The delivery plasmid backbone was amplified using primers vr448 and vr449. The two PCR products were ligated by Gibson Assembly to create plasmid P10.

To create ID Salmonella that express the central domain of NIPP1 (NIPP1-CD, Accession NM_174582), *NIPP1-CD-myc* was cloned into plasmid pLacGFP. *NIPP1-CD-myc* and the backbone plasmid were amplified using primers nd13-nd16 ligated by Gibson Assembly. The *pLac-NIPP1-CD* circuit was cloned using primers nd11 and nd17 (Supplementary Table 4) and inserted into P7 using SacI to create plasmid P11.

To create ID Salmonella that intracellularly deliver CT caspase-3 (*CT Casp-3*, Accession AY219866), parental Salmonella were transformed with plasmid P12. This plasmid was created by PCR amplifying template DNA encoding for CT caspase-3 using primers, vr450 and vr451 from the constitutively two-chain (CT) caspase-3 encoding plasmid pC3D175CT. The pC3D175CT plasmid (Hardy Lab DNA archive Box 7, line 62) was constructed similarly to the caspase-6 CT expression construct[65] using Quikchange mutagenesis on a construct encoding full-length human caspase-3 in a pET23 expression vector (*Addgene*). Plasmid pC3D175CT encodes human caspase-3 residues 1–175, followed by a TAA stop codon, a ribosome binding sequence and the coding sequence for a start methionine and an inserted serine followed by the coding sequence for residues 176–286 with a six histidine tag appended. The backbone of plasmid P8 was PCR amplified using primers vr448 and vr449 and the PCR products were ligated as described above.

**Invasion assays and immunocytochemistry**. Mouse 4T1 or human MCF7 cells were grown on coverslips for fixed-cell imaging or on well plates for live-cell imaging. For fixed imaging, Salmonella were added to 4T1 cultures at a multiplicity of infection (MOI) of 10. For live-cell imaging, Salmonella added to MCF7 cultures at an MOI of 25. The bacteria were allowed to infect cells for two hours. The cultures were then washed five times and treated with 50 μg/ml gentamicin in culture medium to remove extracellular bacteria. Live cells on well plates were directly imaged microscopically.

To obtain detailed images, cells on coverslips were fixed with 10% formalin after 6 or 24 h of incubation. Fixed coverslips were blocked with staining buffer (PBS with 0.1% Tween 20, 1 mM EDTA, and 2% bovine serum albumin) for 30 min. The Tween 20 in this buffer selectively permeabilizes mammalian cell membranes, while leaving bacterial membranes intact. After permeabilization, coverslips were stained to identify Salmonella, released GFP, vacuolar membranes and/or intracellular f-actin with (1) rabbit anti-Salmonella polyclonal antibody (*Abcam*, catalog # ab35156; 1:200 dilution) or FITC-conjugated rabbit anti-Salmonella polyclonal antibody (*Abcam*, catalog # ab69253; 1:100 dilution) (2) rat anti-myc monoclonal antibody (*Chromotek*, catalog # 9e1-100; 1:200 dilution), (3) rabbit anti-LAMP1 polyclonal antibody (*Abcam*, catalog # ab24170; 1:200 dilution), and (4) Alexaflor-568-conjugated phalloidin (*ThermoFisher*, catalog # A12380), respectively.

**Immunohistochemistry**. Excised tumor sections were fixed in 10% formalin, embedded in paraffin and cut into 5 μm sections. Antigen retrieval was performed by incubating deparaffinized sections in 20 mM sodium citrate (pH 7.6) buffer for 20 min at 95 °C. Samples were rehydrated with DI water and Tris buffered saline with 0.1% Tween 20 (TBS-T). Prior to staining, tissue sections were blocked with *Dako* blocking buffer (*Dako*, catalog # X0909). Tissue sections were stained to identify Salmonella and GFP with (1) FITC-conjugated rabbit anti-Salmonella polyclonal antibody (*Abcam*, catalog # ab69253; 1:100 dilution), and (2) either rat anti-*myc* monoclonal antibody (*Chromotek*, catalog # 9e1-100; 1:100 dilution) or rat anti-GFP monoclonal antibody (*Chromotek*, catalog # 3h9-100; 1:100 dilution). Sections were incubated with Alexaflor-568 goat anti-rat secondary antibodies

(*ThermoFisher*, catalog # A11077) and counterstained with DAPI-containing mountant (*ThermoFisher*, catalog # P36962).

**Microscopy**. Samples were imaged on a Zeiss Axio Observer Z.1 microscope. Fixed cells on coverslips were imaged with a ×100 oil immersion objective (1.4 NA). Tumor sections were imaged with ×10 and ×20 objectives (0.3 and 0.4 NA, respectively). Time-lapse fluorescence microscopy of live cells in well plates and tumor-chip devices were housed in a humidified, 37 °C environment and imaged with ×5, ×10, ×63, or ×100 objectives (0.2, 0.3, 1.4, and 1.4 NA, respectively). Fluorescence images were acquired with either 480/525 or 525/590 excitation/emission filters. All images were background subtracted and contrast was uniformly enhanced. Some image analysis was automated using computational code (MATLAB, *Mathworks*).

**Tumor masses in microfluidic devices**. Microfluidic tumor-on-a-chip devices were developed in our laboratory to quantify bacterial invasion[52,66]. Soft lithography was used to create a multilayer device with 12 tumor chambers. Master chips were constructed by spin coating a layer of SU-8 2050 onto a silicon wafer at 1250 RPM for 1 min for a thickness of 150 µm. The silicon wafer with an overlaid mask printed with the microfluidic designs was exposed to UV light (22 J/cm²) for 22 sec. After baking, wafers were developed in PGMEA developing solution for 10 min. PDMS (Sylgard 184) at ratios of 9:1 and 15:1 were used for the channel and valve layers, respectively. After aligning, the layers were baked for 1 h at 95 °C in order to covalently bind. The PDMS device was adhered to a glass slide by placing both in a plasma cleaner (*Harrick*) for 2.5 min. Prior to use, devices were sterilized with 10% bleach and 70% ethanol, and equilibrated with media (DMEM with 20 mM HEPES). Valve actuation was used to position tumor spheroids in the tumor chambers.

**Intracellular Salmonella in cells and tumors**. To observe invasion into cancer cells, Salmonella were administered to mouse 4T1 breast cancer cells on coverslips using an invasion assay. The cells and bacteria were stained with phalloidin and anti-Salmonella antibodies and imaged with ×100 oil immersion microscopy. To measure invasion into cells in tumors, BALB/c mice with subcutaneous 4T1 tumors were administered 2 × 10⁶ CFU of *PsseJ-GFP* Salmonella (Supplementary Table 1) by intratumoral injection. Ninety-six hours after injection, tumors were excised and stained to identify Salmonella and the GFP reporter produced by intracellular Salmonella. The fraction of intracellular Salmonella was determined by identifying Salmonella ($n = 1258$) in four images and determining the number that colocalize with GFP.

**Effect of flhDC on invasion into cells and tumor masses**. To determine the effect of expressing *flhDC* on invasion, 4T1 cells were grown on glass coverslips and administered *flhDC+* and *flhDC−* Salmonella at an MOI of 10. Prior to administration, *flhDC+* Salmonella were grown in LB with 20 mM arabinose to induce *flhDC* expression. For *flhDC+* bacteria, 20 mM arabinose was also added to the mammalian co-cultures to maintain gene expression. Control (*flhDC−*) bacteria were grown without arabinose. Eighteen hours after invasion, the cells were stained to identify intracellular Salmonella. Invasion was quantified in six images from three coverslips per condition by randomly identifying 20 cancer cells from the DAPI channel and determining if there was Salmonella staining within 10 µm of the nucleus.

To quantify invasion into tumor masses, *flhDC*-inducible, intracellular-reporting Salmonella (Supplementary Table 1) were administered to tumor-on-a-chip devices. Bacteria-containing medium (DMEM with 20 mM HEPES) was perfused through the devices for one hour at 3 µm/min for a total delivery of 2 × 10⁶ CFU per device. Two conditions (*flhDC+* and *flhDC−*; $n = 6$ chambers each) were compared. Similar to monolayer culture, *flhDC+* Salmonella were grown in LB with 20 mM arabinose prior to administration, and 20 mM arabinose was added to the co-culture medium to maintain gene expression. Bacterial administration was followed by bacteria-free media (with 20 mM HEPES) for 48 h. Devices were imaged at 30 min intervals. Invasion was quantified at 31 h by measuring GFP expression by invaded bacteria.

**Design of ID Salmonella**. To determine the intracellular activation of the *PsifA* and *PsseJ* promoters, Salmonella with GFP-reporting constructs (Supplementary Table 1) were administered to MCF7 cancer cells at an MOI of 25. Extracellular promoter activity was determined as the average fluorescence intensity of bacteria from three wells, and normalized to the average intensity of *PsseJ* bacteria. The increase in promoter activity following invasion was determined by comparing the average intensity of bacteria in cells to extracellular bacteria. To determine bacterial death caused by lysin E, Salmonella strain *PBAD-LysE* (Supplementary Table 1) was grown in LB to an OD of 0.25 and induced with 10 mM arabinose. Growth and death rates were determined by fitting exponential functions to bacterial density.

To visualize and quantify triggered intracellular lysis and GFP delivery, ID Salmonella were administered to MCF7 cancer cells at an MOI of 25. Cultures were washed five times and treated with 50 µg/ml gentamicin to remove extracellular bacteria. Transmitted and fluorescent images were acquired at ×20 every 30 min for 10 h. Two hundred cancer cells were randomly selected and

scored based on bacterial invasion and lysis. Times of lysis for individual bacteria (within the cancer cells) were determined as the moment of disappearance from the fluorescent time-lapse images of intracellular GFP-expressing bacteria. The lysis fraction was the number of cancer cells with lysed bacteria over the total number of observed cells. The rate of intracellular lysis was determined by fitting an exponential function to the cumulative fraction of cells with lysed bacteria. To generate images of bacterial lysis and GFP delivery, ID Salmonella were administered to 4T1 cancer cells at an MOI of 10. Coverslips were fixed and stained for Salmonella and released GFP. To quantify bacterial protein content, ID Salmonella were suspended at four densities: 10⁶, 10⁷, 10⁸, and 10⁹ bacteria and compared to a GFP standard at three concentrations: 1, 10, and 100 ng per 40 µl Laemmli buffer. GFP was identified in immunoblots with rat anti-GFP monoclonal antibody (*Chromotek*, catalog # 3h9-100; 1:1000 dilution).

**Delivery to tumors**. To identify and quantify GFP delivery to tumor cells, BALB/c mice with subcutaneous 4T1 tumors were administered 2 × 10⁶ CFU of ID Salmonella by intratumoral injection. Ninety-six hours after bacterial injection, tumors, liver and spleens were excised. Tumor sections were stained with anti-GFP antibody (*Abcam*, catalog # ab6556; 1:100 dilution). To compare the amount of delivered protein in the organs of these mice, tumors, livers and spleens were snap-frozen in liquid nitrogen and treated with a buffer containing 50 mM Tris-HCl, 0.3% Triton-X 100, 0.1% NP-40 and 0.3 M NaCl to lyse mammalian cells but not bacterial membranes. Immunoblotting was performed with anti-GPF (*Abcam*, catalog # ab6673; 1:1000 dilution) and anti-β-actin (*GeneTex*, catalog # GTX26276, clone AC-15; 1:1000 dilution). To quantify the amount of protein delivered to tumors, the tumor lysates were run on a similar immunoblot and compared to a GFP standard at 0.43, 1.3, and 3.9 pmols. The amount of GFP per tumor was determined as the lysate concentration multiplied by the lysate volume, normalized by the tumor mass.

To measure the delivery of anti-actin nanobodies, NB and ID Salmonella were administered to 4T1 cancer cells at an MOI of 10. The extent of binding to β-actin was determined by immunoprecipitation. Twenty-four hours after invasion, cells were harvested and centrifuged at $600 \times g$ for 10 min. The cell pellet was lysed, homogenized, and incubated overnight with 50 µl of anti-FLAG purification resin (*Biolegend*, catalog # 651502). Beads were boiled for 5 min and loaded onto SDS-PAGE gels. β-actin was identified with mouse anti-actin monoclonal antibody (*Cell Signaling Technology*, catalog # 8H10D10; 1:1000 dilution).

**Protein release from Salmonella and SCVs**. In order to quantify GFP release from vacuoles, ID Salmonella were administered to 4T1 cancer cells on coverslips at an MOI of 10. At 6 and 24 h, one set of coverslips were fixed, permeabilized and stained with using anti-Salmonella, anti-*myc*, and anti-LAMP1 antibodies. Acquired images were analyzed to quantify (1) the location of released GFP and (2) the location of Salmonella lysis. The fraction of vacuolar GFP was determined as the area of released GFP that was colocalized with LAMP1, normalized by the total area of released GFP. The location of bacteria lysis was determined by identifying all bacteria in seven 86.7 × 66.0 µm regions based on anti-Salmonella staining. Lysed bacteria were determined as those that colocalized with LAMP1. Each lysed bacterium was classified as either vacuolar or cytoplasmic by its colocalization with LAMP1. To visualize the localization of released GFP in cells, a second set of fixed coverslips were stained with anti-Salmonella and anti-*myc* antibodies, and phalloidin, to visualize cell structures and boundaries.

To measure the rate of GFP dispersion through cells after lysis, MCF7 cancer cells were grown on 96-well plates with coverslip glass bottoms (*ThermoFisher*, catalog #160376). ID Salmonella were administered at an MOI of 25. After removing extracellular bacteria with gentamycin, transmitted and fluorescence images were acquired at ×63 every minute for 14 h. Intensities were measured on lines passing through bacterial centers starting when bacteria were intact until diffusion was complete. Cytosolic diffusivity was determined by fitting the spatiotemporal intensity profiles to the radial diffusion equation.

To determine the dependence of protein release on residence in SCVs, 4T1 cancer cells were grown on coverslips and infected with Δ*sifA*, Δ*sseJ*, or ID Salmonella at an MOI of 10 ($n = 3$ for each condition). All three of these strains contained the *PsseJ-LysE* and *Plac-GFP-myc* gene circuits (Supplementary Table 1). At 6 h after invasion, the cancer cells were fixed, permeabilized and stained for Salmonella and released GFP. The lysis fraction was calculated in MATLAB as number of lysis pixels (GFP positive) divided by the total (GFP or Salmonella positive).

**Control of invasion**. To determine the dependence of protein delivery on invasion and intracellular lysis, *flhDC* Sal (contains *PBAD-flhDC*, but not *PsseJ-LysE*) and IDf+ Sal (contains *PBAD-flhDC* and *PsseJ-LysE*; Supplementary Table 1) were administered to 4T1 cancer cells in well plates and on coverslips infected at an MOI of 10. Prior to invasion, *flhDC* expression was induced in the *flhDC+* cultures with 20 mM arabinose. Expression was maintained in the subsequent mammalian co-cultures with addition of 20 mM arabinose. Control (*flhDC-*) bacteria were grown without arabinose. For flow cytometry, cells were processed into a single-cell suspension with 0.05% trypsin (*ThermoFisher*, catalog # 25300-054) and fixed with

5% formaldehyde in PBS and 1 mM EDTA. After permeabilization with 0.1% Tween, cells were stained with FITC-conjugated anti-Salmonella antibody (*Abcam*, catalog # ab69253; 1:100 dilution), and anti-*myc* monoclonal antibody (*Chromotek*, catalog # 9e1-100; 1:100 dilution), followed by Dylight 755 secondary antibody (*Thermofisher*, # SA5-10031; 1:100 dilution) staining against the primary anti-myc antibody. Fluorescence minus one (FMO) of each sample were used as gating controls for each fluorophore. Samples were analyzed on a custom-built flow cytometer (dual LSRFortessa 5-laser, *BD;* with BD FACS Diva software). All fluorophores were compensated with compensation beads (*BD*, catalog # 552845) and did not carry more than 2% bleed over into any other channel. Cells were gated to exclude debris, isolate single cells, and quantify the percentage with intracellular Salmonella and delivered protein (Supplementary Fig. 7). For microscopy, coverslips were fixed, permeabilized and stained for released GFP. Protein (GFP) delivery was quantified in MATLAB as the number of pixels stained for GFP-myc normalized by total in the *PsseJ-LysE−*, *flhDC−* condition.

To determine the effect of *flhDC* on protein delivery, BALB/c mice with subcutaneous 4T1 tumors were injected with $2 \times 10^6$ CFU of IDf+ Salmonella (*flhDC+*) and Δ*flhD* Salmonella (*flhDC−*) via the tail vein. Prior to injection, Salmonella were grown without arabinose to prevent *flhDC* expression until after tumor colonization. At 48 and 72 h after bacterial injection, 100 μg of arabinose in 400 μl of PBS was injected intraperitoneally (IP) into *flhDC+* mice to induce expression. Ninety-six hours after injection, tumors were excised, sectioned and stained with primary anti-GFP antibody (*Chromotek*, catalog # 3h9-100; 1:100 dilution) and followed by anti-rat secondary (*Life Technologies*, catalog # A11077). Delivery was quantified at 20 random points in the transition zone of each acquired tumor image. A point was scored as positive if a cell within 20 μm contained delivered GFP.

**Temporal colonization, biodistribution, and toxicity of ID Salmonella**. To determine the bacterial density in tumors over time, $2 \times 10^7$ CFU ID Salmonella that express firefly luciferase (ID Sal-luc, Supplementary Table 1) were intravenously injected into BALB/c mice with orthotopic 4T1 tumors in the mammary fat pad. At 24, 48, 72, 168, 336 h after bacterial injection, mice were injected IP with 100 μl of 30 mg/ml D-luciferin in sterile PBS and imaged with an IVIS animal imager (*PerkinElmer*, SpectrumCT). Bacterial density was determined as the photon flux. After acquiring the final image at 14 days, the bacterial density was measured by excising tumors and culturing the homogenized tissue on agar plates.

To measure the biodistribution, tumor-free BALB/c mice were injected with $1 \times 10^7$ ID Salmonella via the tail vein. Control mice were injected with sterile saline. After 14 days, six organs were excised and weighed: spleen, liver, lung, kidney, heart, and brain. Organs were minced and cultured on agar plates. A second experiment was performed to determine the biodistribution at earlier times. Tumor-free C57L/J mice were intravenously injected with either saline (control), $1 \times 10^7$ ID Salmonella, or $1 \times 10^7$ CT Casp-3 Salmonella. At two timepoints (6 h and 7 days), tissues were excised and weighed from two separate groups of mice. Bacterial densities were determined by mincing organs and culturing on agar plates.

To measure the toxicity of ID Salmonella, four tumor-free BALB/c mice were injected with $1 \times 10^7$ ID Salmonella via the tail vein. After 14 days, whole blood was isolated by percutaneous cardiac puncture. A second experiment was performed to determine the toxicity of bacterial delivery of CT Casp-3. Tumor-free C57L/J mice were intravenously injected with saline (control), $1 \times 10^7$ ID Salmonella, or $1 \times 10^7$ CT Casp-3 Salmonella. Sera were collected 7 days after injection. Chemistry profiling and comprehensive hematology was conducted on the serum and whole blood samples by IDEXX Laboratories (Grafton, MA).

**Cytotoxicity of CT Casp-3 and NIPP1-CD**. To measure the efficacy of delivering protein drugs, *NIPP1-CD* and *CT Casp-3* Salmonella were administered to Hepa 1–6 liver cancer cells at an MOI of 10. Cell death was detected with 500 ng/ml ethidium homodimer and calculated as the fraction of dead Salmonella-invaded cells over the total number of Salmonella-invaded cells. To measure cell death in tumor masses, media containing $2 \times 10^7$ CFU/ml *NIPP1-CD* or *CT Casp-3* Salmonella and 500 ng/ml ethidium homodimer was perfused through tumor-on-a-chip devices for 1 h at 3 μm/min. Bacterial administration was followed by bacteria-free media. Transmitted and fluorescence images were acquired every 30 min for 24 h. Death was quantified as the percentage of the tumor mass stained with ethidium homodimer at 24 h.

To measure efficacy and the extent of delivering NIPP1-CD, $1 \times 10^7$ CFU/mouse of *NIPP1-CD* Salmonella or saline (controls) were administered to BALB/c mice with subcutaneous 4T1 tumors by intravenous injection. Tumors were measured twice a week and volumes were calculated with the formula (length) *(width$^2$)/2. After 31 days, tumors were excised and stained for Salmonella (*Abcam*, catalog # ab69253; 1:100 dilution) and NIPP1-CD with antibodies to the c-terminal *myc* tag (*Chromotek*, catalog # 9e1-100; 1:100 dilution), followed by a secondary antibody to the *myc* specific antibody (*Life Technologies*, catalog # A11077; 1:100 dilution). In the acquired images, DAPI staining was used to identify regions with viable nucleated cells. The average delivery of NIPP1 was determined as the fraction of the viable cell area that positively stained for delivered NIPP1-CD.

To measure the efficacy of delivering CT Casp-3, bacteria were administered to BALB/c mice with subcutaneous 4T1 tumors, BALB/c mice with subcutaneous BNL-MEA tumors, and C57L/J mice with subcutaneous Hepa 1–6 tumors. For the 4T1 and Hepa 1–6 tumor models, groups of mice ($n = 6$ for 4T1; $n = 3$ for Hepa 1–6) received intratumoral injections of saline, $4 \times 10^7$ CFU ID Salmonella, or $4 \times 10^7$ CFU *CT Casp-3* Salmonella. The ID Salmonella control established the baseline effect of bacteria colonization and intracellular lysis. Every 5 days, tumors were injected with bacteria or saline. For the BNL-MEA model, once tumors reached 100 mm$^3$, mice were intravenously injected every five days with $1 \times 10^7$ CFU/mouse of *CT Casp-3* Salmonella ($n = 5$), 10 mg/kg Sorafenib ($n = 4$), or saline ($n = 5$). Sorafenib (10 mg/kg) is the standard-of-care for liver cancer. Tumors were measured twice a week and volumes were calculated with the formula (length)*(width$^2$)/2. Mice were sacrificed when tumors reached 1000 mm$^3$.

To measure the efficacy of CT Casp-3 in secondary tumor sites, lung metastases were formed by injection of $5.0 \times 10^4$ luciferase-expressing 4T1 cells into the tail veins of female BALB/c mice. Relative metastasis volume was determined by injecting the mice IP with 100 μl of 30 mg/ml D-luciferin in sterile PBS and imaged with an IVIS animal imager (*PerkinElmer*, SpectrumCT). When lung colonization was detected, mice were injected intravenously every 5 days with $1 \times 10^7$ CFU CT Casp-3 ID Salmonella or injected IP with 10 mg/kg paclitaxel. Tumor burden was monitored weekly with BLI until study endpoint.

**Statistical methods**. Statistical analysis was performed in Excel (Microsoft Office Professional Plus 2016) and GraphPad Prism 9.2.0. Comparisons of two populations were made with two-tailed, unpaired Student's *t*-tests. Comparisons of multiple conditions were made using ANOVA with a Bonferroni correction. Comparison of multiple conditions to a single control were performed with ANOVA followed by Dunnett's method. In some large datasets, outliers were removed using the ROUT method, with a Q (maximum false discovery rate) of 1%. To compare survival, log-rank tests were used with Bonferroni correction. All measurements were taken from distinct samples. Values are reported as means ± standard errors (SEMs). Statistical significance was confirmed when $P < 0.05$.

**Reporting summary**. Further information on research design is available in the Nature Research Reporting Summary linked to this article.

## Data availability

All data generated in this study are available within the Article, Supplementary Information or Source Data file. Source data generated in this study have been also deposited in the *figshare* database with digital identifier https://doi.org/10.6084/m9.figshare.16439073 [67]. Source data are provided with this paper.

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

## Acknowledgements

We gratefully acknowledge financial support from the National Cancer Institute of the National Institutes of Health, grants R01CA188382 (to A.V.E. and N.S.F.), R43CA250941

(to N.V.D. and N.S.F.), and R43CA9622551 (to N.V.D. and N.S.F.); the National Science Foundation, grants 1819794 (to N.V.D. and N.S.F.) and 2035560 (to N.V.D. and N.S.F.); the Department of Defense, grant W81XWH1910602 (to N.S.F.); and the Manning/IALS Innovation Award from UMass Amherst.

## Author contributions

V.R., N.V.D. and N.S.F. designed the study; V.R., N.V.D., C.H., V.W., S.W., E.K., S.B. and A.S. performed experiments; V.R., N.V.D., C.H. and N.S.F. collected and analyzed data; J.H. provided CT Casp-3; M.B. and A.V.E. provided NIPP1-CD; J.H., M.B. and A.V.E. gave technical support and conceptual advice; V.R., N.V.D. and N.S.F. wrote the manuscript.

## Competing interests

The authors declare the following competing interests: V.R., N.V.D., and N.S.F. are founders of Ernest Pharmaceuticals, LLC. The remaining authors declare no competing interests.
