## [Peer Review File · Nature Communications]

REVIEWER COMMENTS

Reviewer #1 (Remarks to the Author):

The manuscript describes an effective intracellular delivery system by non-pathogenic Salmonella (ID Salmonella) characterized with three genetic circuits for synthesis, invasion, and release. The invasion and survival machinery of Salmonella could express protein within cancer cells in solid tumors. The lysis system could control protein release into SCVs and then diffuse throughout the cytoplasm. ID Salmonella could provide a controllable system to limit the release of protein inside cells and demonstrated the delivery of CT Casp-3 that retarded the growth of lung metastases and hepatocellular carcinoma and increased the survival of the treated mice. Despite the characteristics of enhanced invasion and intracellularly synthesis and release of protein drugs, the reviewer has a major concern regarding the specificity to cancer cells as these could also occur in healthy cells, such as normal immune cells, hepatocytes and splenocytes. Although the authors showed the main accumulation of ID Salmonella in tumor 2 weeks post-treatment, which was resulted from the proliferation, the majority of the dosed bacteria would accumulate in liver, spleen as well as other major organs upon injection. Once internalization into normal cells, the synthesis, invasion, and release associated with ID Salmonella could happen similarly. Therefore, the authors oversold the work by claiming "deliver protein payloads specifically into cancer cells", "release proteins exclusively in tumor cells" and "is safe and self-limiting". In addition, as ID Salmonella only retarded tumor growth, the use of "eliminate solid tumors" and similar descriptions should be avoided. The manuscript should be revised appropriately before consideration for publication.

Major comments:

1. To evaluate the safety, ID Salmonella with NIPP1-CD or CT casp-3 rather than GFP should be performed as the safety issues is mainly associated with the payloads.
2. To understand the acute toxicity, biodistribution of ID Salmonella should be carried out within shorter periods in healthy mice as well as tumor-bearing mice, such as hours to a few days post-injection. Additionally, this could show the accurate distribution of the injected bacteria.
3. A group of three mice is too small for tumor growth and survival studies.
4. The tumor density initially increased and then decreased, reaching a peak at 72 h (Figure 5A-B). Proper explanations should be added.

Reviewer #2 (Remarks to the Author):

The manuscript entitled “Intracellular delivery of protein drugs with bacteria designed to invade and autonomously lyse: a new tool to eliminate solid tumors” describes a genetic engineering of Salmonella strain with genetic circuit that activate cancer cell invasion and release protein drugs through lysis exclusively in tumor cells. This study is interesting and contains intriguing technologies that may be able to improve anticancer therapeutics. However, some points need to be clarified and supplemented before publication.

1. First of all, in this study, the nature of genetically engineered bacterial strains should be clearly characterized. The parental strain is VNP20009. Fig 1's strain flhDC (-) strain would be generated based on VNP 20009? And re-express flhDC in flhDC (+) strain? ID Salmonella seems to be transformed with P_{ssE}-LysE, but not with PBAD-flhDC. The ID Sal would just have natural flhDC of parental strain? In Fig. 3, what's difference between ID Sal and P_{ssE}? ID Sal is the strain based on VNP20009 and P_{ssE} is based on P_{ssE} VNP20009? But in Fig. 4 ID Salmonella is described to be transformed with P_{ssE}-LysE and PBAD-flhDC. Then, this strain is the one overexpress flhDC based on VNP20009? VNP20009 should have their intrinsic flhDC, then why flhDC (-) cannot invade cells? Many things related to strain are confusing and unclear. One more, therapeutic strains (NIPP1-CD and CT Cas-3) has no additional flhDC gene induction. Then this should be based on VNP20009 that has intrinsic flhDC. Does this strain can invade cancer cells?
2. Figure legends summarized the results that is repeated in the Result section. But they are lack of information of the experiments such as number of animals, days after treatment, some important materials, etc. Authors need to re-edit figure legends to give this information.
3. In vivo therapeutic efficacy was evaluated in 4T1 and Hepa 1-6 tumor models using CT Casp-3 bacteria. But NIPP1-CD was not evaluated in terms of therapeutic efficacy in mouse models. It is highly recommended to include NIPP1-CD data, otherwise, this study looks like preliminary study. Moreover, CT-Casp-3 was evaluated in very small group of animals: n=6 for 4T1, n=3 for Hepa1-6. Authors should perform this experiment with larger number of animals.
4. Line 112: how you can calculate half-life of 2.1 h and 163,000 GFP molecules. This should be described in Result section.
5. Fig. 3 I: Cytosolic bacteria were stained with light blue and did not lyse. How are you sure of this? Why are you convinced that GFP did not come from these cytosolic bacteria?
6. Fig 4B: Parental strain will also have FlhDC, but why can't they invade cells?
7. Line 168: How did you measure GFP amount (60±12 ug/g tumor)?
8. Fig. 5C-E: What dpi (days post-inoculation of bacteria) you measured these data? Bacterial distribution differs according to dpi. Initially high in spleen and liver, but later high in tumor. Authors should make this clear.
9. Line 183: The tumor density -> This seems to be “the bacterial density in tumor”?

10. Authors did not provide accurate information of luciferase. Is it firefly or gaussian luciferase? How was D-luciferin injected (dose, concentration). There is no information about L-arabinose use (dose for in vivo or in vitro use).

11. Fig. 6C: This result is not clear whether it was measured in animal models or other in vitro model. Authors should clarify it.

12. Fig. 7F: There is no information about each line. Red line seems to be CT-Casp-3. Open triangle is paclitaxel? Then, what is dotted line?

Reviewer #3 (Remarks to the Author):

In the manuscript "Intracellular delivery of protein drugs with bacteria designed to invade and autonomously lyse: a new tool to eliminate solid tumors" Raman et al. described novel, Salmonella-based intracellular drug delivery system to tumor cells. To successfully deliver protein drugs to tumor cells in vitro and in vivo, they engineered Salmonella strain to invade, lyse and release proteins in the cytoplasm of exclusively tumor cells. In the study, they predominantly used 4T1, mouse breast cancer cell line, Hepa1-6, mouse liver cancer cell line, microfluidic tumor masses which mimic tumor tissue bordering blood vessel, and mouse model of breast and liver cancer. Using Salmonella-based system to deliver constitutively active caspase 3 (and to a lesser extent NIPP1-CD), they provided evidence that this approach can be viable option for delivery of different protein drugs for the treatment of solid tumors.

Efficient drug delivery specifically to cancer cells is one of the major challenges in cancer therapy. Hence, this is a very interesting study, with enormous clinical interest and potential; however, it is not suitable for publication in its presented form. Additional experiments, controls and clarifications are needed for manuscript to be accepted for publication.

General concerns:

1. Mouse experiments are not appropriately explained. Additional clarification is needed. Are Hepa1-6 cells injected subcutaneously or in the liver? Is this xenograft or orthotopic mouse model? Please explain for all experiments.

2. Description of some mouse experiments in the Materials and methods section states that 4T1 cells are transplanted into mammary fat pad (orthotopic model). Does that apply to every experiment?

3. Were bacteria injected into the mice always through the tail vein?

4. The number of animals used per treatment should be clearly written in each figure.
5. Labeling should be constant through the manuscript. For example: in the Fig2., it is written Pssel-LysE, but in the Legend of the Fig2. It is written PsseJ-lysE.
6. Adding color code in the figure, not only Figure legend, would make it easier for readers.

Specific comments:

1. Fig4E.- How was bacteria delivered into the mice? What about expression of GFP in the lungs (Fig.4F.)?
2. Is the drug delivery to 20% of tumor cells significant (Fig4I)? How long after bacterial injection tumors were harvested?
3. Fig5C., some bacterial infiltration has been observed in the liver. Did the authors look for histological signs of liver inflammation/infiltration of immune cells? How long after bacterial injection was blood liver panel done (Fig5E)?
4. Was the experiment with NIPP1-CD (Figure 5) performed in 4T1 cells as well (in vitro and in vivo)?
5. In the Fig6D successful delivery of NIPP1-CD to tumor cells was observed. Was there any effect on tumor cells? Apoptosis? It would be informative if delivery to the tumor cells is quantified (percentage of tumor cells with delivered NIPP1-CD).
6. Figure 7E, what about metastasis?
7. Figure 7C, there is significant cell death observed even in the tumor masses treated only with control. Can authors comment on that?
8. Please clarify if in the Figure 7G, are metastasis formed in orthotopic model or by cells injected through tail vein?
9. Was safety study (Figure 5) performed with Salmonella strains that release caspase3 and NIPP1-CD? Because some bacteria colonize liver and spleen, releasing these proteins could potentially induce damage.
10. Was there any toxicity observed in mice treated long-term with bacteria?

Response to Reviewers' Comments

Intracellular delivery of protein drugs with bacteria designed to invade and autonomously lyse: a new tool to reduce solid tumors

Vishnu Raman^{*,1,2}, Nele Van Dessel^{*,2}, Christopher Hall¹, Victoria Wetherby², Samantha Whitney¹, Emily Kolewe¹, Shoshana Bloom¹, Abhinav Sharma¹, Jeanne Hardy^{3,4,5}, Mathieu Bollen⁶, Aleyde Van Eynde⁶, Neil S. Forbes^{1,2,4,5,†}

Submitted for publication in *Nature Communications*
Manuscript ID: NCOMMS-20-34783-T

We would like to thank the reviewers for their thoughtful comments, which have all been addressed. The responses and modifications incorporated in the revision are listed below. Text that was altered from the original manuscript is indicated in red. The locations of edits are specified in each response.

Reviewer #1

The manuscript describes an effective intracellular delivery system by non-pathogenic *Salmonella* (ID *Salmonella*) characterized with three genetic circuits for synthesis, invasion, and release. The invasion and survival machinery of *Salmonella* could express protein within cancer cells in solid tumors. The lysis system could control protein release into SCVs and then diffuse throughout the cytoplasm. ID *Salmonella* could provide a controllable system to limit the release of protein inside cells and demonstrated the delivery of CT Casp-3 that retarded the growth of lung metastases and hepatocellular carcinoma and increased the survival of the treated mice. Despite the characteristics of enhanced invasion and intracellularly synthesis and release of protein drugs, the reviewer has a major concern regarding the specificity to cancer cells as these could also occur in healthy cells, such as normal immune cells, hepatocytes and splenocytes. Although the authors showed the main accumulation of ID *Salmonella* in tumor 2 weeks post-treatment, which was resulted from the proliferation, the majority of the dosed bacteria would accumulate in liver, spleen as well as other major organs upon injection. Once internalization into normal cells, the synthesis, invasion, and release associated with ID *Salmonella* could happen similarly. Therefore, the authors oversold the work by claiming "deliver protein payloads specifically into cancer cells", "release proteins exclusively in tumor cells" and "is safe and self-limiting". In addition, as ID *Salmonella* only retarded tumor growth, the use of "eliminate solid tumors" and similar descriptions should be avoided. The manuscript should be revised appropriately before consideration for publication.

Response: The reviewer raises important concerns about toxicity to normal cells and language in the manuscript. To address the concern about toxicity, we performed additional experiments to (1) measure the biodistribution of caspase-delivering *Salmonella* at earlier time points (6 hr and 7 days) and (2) measure the toxicity of caspase-delivering bacteria. The results from these experiments are included in Supplemental Figures S3 and S4. They show that, as the reviewer suggested, bacteria are present in the liver and spleen soon after injection (Figure S3). This result was expected because these are the major clearance organs for *Salmonella*. By 7 days, bacteria had cleared from most organs (Figure S3B). In mice, these bacteria did not cause any adverse

health effects and no toxicities were detected in the blood (Figure S4). To address this concern, we added text to the Results (lines 228-231), the Discussion (lines 314-320) and the Supplemental Results (Figures S3 & S4).

As suggested by the reviewer, we have also changed language throughout the manuscript. We have removed the word “eliminated” from the Title, the Abstract (lines 9-10), the caption title for Figure 7, and the Discussion (line 250). We have also made all of the suggested changes to remove claims of exclusive delivery to cancer cells. Text was changed in the Abstract (line 5), the Introduction (lines 67-68), the Results (line 192), and the Discussion (line 315).

Major comments:

1. To evaluate the safety, ID Salmonella with NIPP1-CD or CT casp-3 rather than GFP should be performed as the safety issues is mainly associated with the payloads.

Response: As requested, we analyzed the safety of ID Salmonella with CT casp-3. We obtained similar results as with ID Salmonella (in Figure 5). At seven days after injection, no difference in toxicity was observed in mice that received *CT Casp-3* Salmonella compared to either bacterial or saline controls. These results were added as Figure S4 in the Supplemental Information and text was added to the Results (lines 228-231) and the Discussion (lines 314-320).

2. To understand the acute toxicity, biodistribution of ID Salmonella should be carried out within shorter periods in healthy mice as well as tumor-bearing mice, such as hours to a few days post-injection. Additionally, this could show the accurate distribution of the injected bacteria.

Response: As requested, we measured the biodistribution of ID Salmonella at earlier times (6 hours and 7 days after injection). At six hours, the bacteria were predominantly in the liver, spleen and kidneys (Figure S3A). This result was expected because these are the major clearance organs for Salmonella. By seven days, the bacteria had cleared from all organs except for trace amounts in the spleen (< 10 CFU/g; Figure S3B). The initial accumulation in these organs did not affect the health of the mice and did not affect the blood chemistry (Figure S4). The clearance of the bacteria between 6 hours and 7 days was caused by the autolysis after invasion (Figure 5B). To address this concern, these results were added to Supplemental Results (Figure S3 & S4), and text was added to the Results (lines 228-231) and the Discussion (lines 314-320).

3. A group of three mice is too small for tumor growth and survival studies.

Response: As requested, we repeated the tumor growth experiments with *CT casp-3* Salmonella with a second liver cancer model. Subcutaneous tumors were formed in BALB/c mice by injection of syngeneic BNL-MEA cells. Once tumors formed, mice were intravenously injected with either 1×10^7 CFU/mouse of *CT Casp-3* Salmonella ($n = 5$), 10 mg/kg Sorafenib ($n = 4$), or saline ($n = 5$). We saw similar results as with the Hepa 1-6 tumors. After ten days, *CT Casp-3* Salmonella reduced tumor volume by about 50% compared to Sorafenib, which is the standard-of-care for liver cancer. To address this concern, we have added the new results as Figure 7H and added text to the caption to Figure 7, the Results (lines 236-240) and the Methods (lines 615-618, and 621-624).

4. The tumor density initially increased and then decreased, reaching a peak at 72 h (Figure 5A-B). Proper explanations should be added.

Response: The peak in bacterial density at 72 h was most likely caused by the timing of invasion and lysis of ID Salmonella after injection into tumor-bearing mice. During the first 72 hours, the injected bacteria most likely accumulated in tumor tissue, invaded into cancer cells, and formed SCVs. Once inside SCVs, the *Pssej-LyE* circuit was triggered and the bacteria lysed, resulting in clearance from the cancer cells and the tumors. This lysis resulted in reduced bacterial densities in the healthy organs compared to previous measurements (Forbes NS, et al. 2003. *Cancer Res.* 63:5188 and Felgner S, et al. 2018. *Oncoimmunology* 7:e1382791). In Felgner et al., the bacterial density in the spleen remained relatively constant for 144 hours. To address this concern, we adjusted the caption for Figure 5 and added a new paragraph (lines 260-268) to the Discussion.

Reviewer #2

The manuscript entitled “Intracellular delivery of protein drugs with bacteria designed to invade and autonomously lyse: a new tool to eliminate solid tumors” describes a genetic engineering of Salmonella strain with genetic circuit that activate cancer cell invasion and release protein drugs through lysis exclusively in tumor cells. This study is interesting and contains intriguing technologies that may be able to improve anticancer therapeutics. However, some points need to be clarified and supplemented before publication.

1. First of all, in this study, the nature of genetically engineered bacterial strains should be clearly characterized. The parental strain is VNP20009. Fig 1's strain *flhDC* (-) strain would be generated based on VNP 20009? And re-express *flhDC* in *flhDC* (+) strain? ID Salmonella seems to be transformed with *PsseJ-LysE*, but not with *PBAD-flhDC*. The ID Sal would just have natural *flhDC* of parental strain? In Fig. 3, what's difference between ID Sal and Δ *sseJ*? ID Sal is the strain based on VNP20009 and Δ *sseJ* is based on Δ *sseJ* VNP20009? But in Fig. 4 ID Salmonella is described to be transformed with *PsseJ-LysE* and *PBAD-flhDC*. Then, this strain is the one overexpress *flhDC* based on VNP20009? VNP20009 should have their intrinsic *flhDC*, then why *flhDC* (-) cannot invade cells? Many things related to strain are confusing and unclear. One more, therapeutic strains (NIPP1-CD and CT Cas-3) has no additional *flhDC* gene induction. Then this should be based on VNP20009 that has intrinsic *flhDC*. Does this strain can invade cancer cells?

Response: We agree with the reviewer that the information about the strains could be presented more clearly. Most of the reviewer's interpretations about the strains are correct. The following are specific answers to the questions:

- a) The parental strain is VNP20009 with an *asd* knockout (i.e. Δ *msbB*, Δ *purl*, Δ *xyl*, Δ *asd*). To address this, text was added to the Results (lines 81-83), the Methods (lines 334-335) and the Supplemental Methods (lines S57-66).
- b) In Figure 1, both conditions (*flhDC*- and *flhDC*+) used the *flhDC* Sal strain (see Table S1), which is Δ *flhD* Salmonella transformed with *PBAD-flhDC*. The *flhDC*- bacteria were not

induced and the *flhDC*⁺ bacteria were induced with 20 mM arabinose. In Salmonella, deletion of *flhD* is sufficient to prevent the function of the FlhDC hetero-oligomeric complex (Wang S. et al. 2006. *J Mol Bio.* 355:798). To address this point, text was changed in the caption for Figure 1, the Results (lines 88-92 and 96) the Methods (lines 336-337) and the Supplemental Methods (lines S67-74).

- c) ID Salmonella is the parental strain transformed with *PsseJ-LysE* and *Plac-GFP*. It does not contain *PBAD-flhDC* and natively expresses *flhDC*. It also naturally invades into epithelial cells (Figure 2C,H). To clarify the nomenclature, the strain with re-expressed *flhDC* (which was transformed with *PsseJ-LysE*, *Plac-GFP*, and *PBAD-flhDC*) was named IDf⁺ Salmonella. As the reviewer points out, the original Figure 4 was confusing because it contained results with both ID and IDf⁺ Salmonella. To make this clearer, the results with ID Salmonella were moved to Figure 2 (new panels H-J). After this change, Figure 2 now focusses on ID Salmonella and Figure 4 focusses on IDf⁺ Salmonella. To address this change and better describe the strains, text was changed in the Results (lines 125-134, 175-178, 181, 185-189), the Methods (lines 540-544 and 560-563), the captions for Figures 2-4, the Supplemental Methods (line S132), and Table S1.
- d) The Δ *sseJ* strain is ID Salmonella with *sseJ* deleted (i.e. it is the *sseJ* knockout transformed with *PsseJ-LysE* and *Plac-GFP*). It delivers GFP but does not leave SCVs. Similarly, the Δ *sifA* strain is ID Salmonella with *sifA* deleted. This strain has the machinery to deliver GFP, but is predominantly cytoplasmic. To clarify the connection to ID Salmonella, these strains have been renamed Δ *sseJ* ID Sal and Δ *sifA* ID Sal. To address these points, a section was added to the Supplemental Methods (lines S127-131 and Table S1) and text was added to the Results (lines 165-170), and the caption for Figure 3.
- e) The therapeutic strains that deliver NIPP1-CD and CT Casp-3 were based on ID Salmonella and were not transformed with *PBAD-flhDC*. To address this point, text was added to the caption for Figure 7, the Results (lines 208, 216) and the Methods (lines 342-344).

2. Figure legends summarized the results that is repeated in the Result section. But they are lack of information of the experiments such as number of animals, days after treatment, some important materials, etc. Authors need to re-edit figure legends to give this information.

Response: As requested, information about the number of animals and the time of treatment was added to the captions for Figures 1, 2, 4, 5, 6 and 7. In response to reviewer #3, information was also added about the location of the tumors and the site of bacterial injection.

3. In vivo therapeutic efficacy was evaluated in 4T1 and Hepa 1-6 tumor models using CT Casp-3 bacteria. But NIPP1-CD was not evaluated in terms of therapeutic efficacy in mouse models. It is highly recommended to include NIPP1-CD data, otherwise, this study looks like preliminary study. Moreover, CT-Casp-3 was evaluated in very small group of animals: n=6 for 4T1, n=3 for Hepa1-6. Authors should perform this experiment with larger number of animals.

Response: To address these concerns we have performed additional experiments and included more data in the manuscript. We repeated the efficacy experiment with a second liver cancer model: subcutaneous BNL-MEA tumors in BALB/c mice. We saw similar results to the experiment with the Hepa 1-6 tumors. After ten days, CT Casp-3 Salmonella reduced tumor volume by about 50% compared to Sorafenib, which is the standard-of-care for liver cancer. To address this

concern, we have we have added the new results as Figure 7H and added text to the caption to Figure 7, the Results (lines 236-239) and the Methods (lines 366-369, 615-617, and 621-624).

We have also added results showing the efficacy of NIPP1-CD Salmonella in tumor-bearing BALB/c mice to the Supplemental information (Figure S2). We included the data with NIPP1 to show that ID Salmonella is capable of delivering multiple types of therapeutic proteins. Although NIPP1-CD caused cell death in cultured cells and tumor masses in vitro, it did not affect tumor volumes in mice. This shows that the efficacy of a bacterial delivered protein is dependent on many factors, more than just cellular cytotoxicity. To address this point, text was added to the Results (lines 213-214), the Discussion (lines 256-258 and 304-309), and the Supplemental Results (Figure S2).

4. Line 112: how you can calculate half-life of 2.1 h and 163,000 GFP molecules. This should be described in Result section.

Response: After invasion, the Salmonella lysed over the course of ten hours. By tracking individual cells, bacterial lysis was determined from the fluorescence time-lapse images as cells that have GFP-containing bacteria that disappear with time. Of a population of cancer cells that had been invaded by Salmonella, we measured the time when the bacteria disappeared. The fraction per hour and the cumulative fraction are reported in Figure 2E. We calculated the lysis rate and the half-life from the cumulative fraction of lysed bacteria. The number of molecules of GFP per bacterium was determined by quantitative immunoblot with known numbers of bacteria compared to known amounts of a protein standard (Figure 2G). To clarify this information, text was added to the Results (lines 116-119 and 121-122), the caption for Figure 2, and the Methods (lines 472-477).

5. Fig. 3 I: Cytosolic bacteria were stained with light blue and did not lyse. How are you sure of this? Why are you convinced that GFP did not come from these cytosolic bacteria?

Response: As the reviewer's question indicates, the description of these results was not clear in the manuscript. Based on two separate experiments, we showed that (1) Salmonella outside of SCVs did not lyse, and (2) released GFP was supplied by lysed bacteria inside SCVs. For the first experiment, lysed bacteria were identified in cells by co-localization of staining for Salmonella and released GFP. More than 95% of these lysed bacteria co-localized with SCVs (LAMP1; Figure 3J). To confirm this result, we transformed $\Delta sifA$ Salmonella, which are predominantly cytoplasmic (Beuzon CR et al. 2000. Embo J 19:3235), with the GFP-delivering machinery of ID Salmonella (i.e. *PsseJ-LysE* and *Plac-GFP*). After application to cancer cells (Figure 3K, *upper left*), these bacteria invaded (*red*), but did lyse or released GFP (i.e. did not stain for GFP). In comparison, both ID Salmonella and $\Delta sseJ$ ID Salmonella (which are predominantly vacuolar), invaded (*red*), lysed and released GFP (*green*). To address this concern, we adjusted the axis labels for Figures 3F and 3J; renamed the strains in Figure 3K as $\Delta sifA$ ID Salmonella and $\Delta sseJ$ ID Salmonella to make it clearer that the strains contain the GFP-delivering machinery; and adding text to the caption for Figure 3, the Results (lines 164-170) and the Methods (lines 513-519).

6. Fig 4B: Parental strain will also have FlhDC, but why can't they invade cells?

Response: This is a good question that shows that clarification is required in the manuscript. The parental strain (*Par*) naturally expresses *flhDC* and does invade into cells. This can be seen in Figures 1B, 2C, 2H, and 3K. The controls used in the *flhDC* experiments in Figures 1 and 4 (*flhDC*-) have had *flhD* deleted and do not invade. To address this concern, we have arranged the figures so that results with *flhDC*+ and *flhDC*- are grouped together in Figure 1 and 4 (see the response to comment 1) and by clarifying that (1) the parental strain expresses native levels of *flhDC* in the Results (lines 90-91) and (2) the *flhDC*- condition uses $\Delta flhD$ Salmonella in the caption for Figures 1 and 4, and the Results (lines 88-89 and 96).

7. Line 168: How did you measure GFP amount (60 ± 12 ug/g tumor)?

Response: The GFP amount was calculated by comparison to a GFP standard. A geometric series of the standard was run on a quantitative immunoblot at 0.43, 1.3 and 3.9 pmols. The amount of GFP per tumor was determined as the measured concentration multiplied by the combined volume of the tumor lysate and the lysis buffer. This amount was normalized by the tumor mass. To address this concern, text was added to the Results (lines 130-131) and the Methods (lines 495-499).

8. Fig. 5C-E: What dpi (days post-inoculation of bacteria) you measured these data? Bacterial distribution differs according to dpi. Initially high in spleen and liver, but later high in tumor. Authors should make this clear.

Response: The measurements in Figures 5C-E were performed 14 days post inoculation. In addition, two additional experiments were performed at earlier times points (6 h and 7 days) to respond to concerns of reviewer 1. The results of these experiments have been added to the Supplemental Information (Figure S3). To address this concern, the timings of the measurements were added to the caption for Figure 5.

9. Line 183: The tumor density -> This seems to be “the bacterial density in tumor”?

Response: The reviewer is correct. To reflect this, we have changed the text as suggested in the Results (line 194) and the Methods (line 570).

10. Authors did not provide accurate information of luciferase. Is it firefly or gaussian luciferase? How was D-luciferin injected (dose, concentration). There is no information about L-arabinose use (dose for in vivo or in vitro use).

Response: Two experiments used bioluminescent imaging: the bacterial clearance experiment in Figure 5A,B and the metastasis volume experiment in Figure 7F,G. Both of these experiments used firefly luciferase. Prior to imaging in both experiments, mice were injected intraperitoneally with 100 μ l of 30 mg/ml D-luciferin in sterile PBS. To address this concern, the information about the dose and concentration of D-luciferin was added to the captions for Figure 5 and 7 and to the Methods (lines 571-573 and 630-631).

Arabinose was used in the experiments in Figures 1 and 4 to induce the expression of *flhDC*. For the cell culture experiments, *flhDC*+ Salmonella were grown in LB with 20 mM arabinose to induce the *PBAD-flhDC* circuit prior to administration. After addition to cultures of cancer cells, the co-culture medium was also supplemented with 20 mM arabinose to maintain expression. For mice studies, bacteria were grown in cultures without arabinose before injection. At 48 and 72 h after bacterial injection, 100 µg of arabinose in 400 µl of PBS was injected IP into *flhDC*+ mice. This timing induced expression after the bacteria had colonized tumors and cleared from healthy organs. To address this concern, this information regarding *flhDC* induction was added to the captions for Figures 1 and 4 and the Methods (lines 442-445, 453-455, 542-544, 560-563).

11. Fig. 6C: This result is not clear whether it was measured in animal models or other in vitro model. Authors should clarify it.

Response: The result in Figure 6C is from the tumor masses shown in Figure 6B. This point has been clarified in the caption for Figure 6.

12. Fig. 7F: There is no information about each line. Red line seems to be CT-Casp-3. Open triangle is paclitaxel? Then, what is dotted line?

Response: The reviewer is correct; the red squares are CT Casp-3 and the open triangles are paclitaxel. The dashed line is eight times the initial volume, based on the bioluminescent flux. Treatment with *CT Casp-3* Salmonella kept growth beneath this limit, whereas metastatic volume increased 85 fold after treatment with Paclitaxel. To address this concern, a legend has been added to Figure 7G, the units have been changed to relative metastatic volume, and the caption for Figure 7 has been updated.

Reviewer #3

In the manuscript “Intracellular delivery of protein drugs with bacteria designed to invade and autonomously lyse: a new tool to eliminate solid tumors” Raman et al. described novel, Salmonella-based intracellular drug delivery system to tumor cells. To successfully deliver protein drugs to tumor cells in vitro and in vivo, they engineered Salmonella strain to invade, lyse and release proteins in the cytoplasm of exclusively tumor cells. In the study, they predominantly used 4T1, mouse breast cancer cell line, Hepa1-6, mouse liver cancer cell line, microfluidic tumor masses which mimic tumor tissue bordering blood vessel, and mouse model of breast and liver cancer. Using Salmonella-based system to deliver constitutively active caspase 3 (and to a lesser extent NIPP1-CD), they provided evidence that this approach can be viable option for delivery of different protein drugs for the treatment of solid tumors.

Efficient drug delivery specifically to cancer cells is one of the major challenges in cancer therapy. Hence, this is a very interesting study, with enormous clinical interest and potential; however, it is not suitable for publication in its presented form. Additional experiments, controls and clarifications are needed for manuscript to be accepted for publication.

General concerns:

1. Mouse experiments are not appropriately explained. Additional clarification is needed. Are Hepa1-6 cells injected subcutaneously or in the liver? Is this xenograft or orthotopic mouse model? Please explain for all experiments.

Response: As requested, additional information has been provided for all mouse experiments. All in vivo experiment were performed in syngeneic C57L/J and BALB/c mice models with complete immune systems. Hepa 1-6 tumors were implanted subcutaneously in C57L/J mice. This information has been added to the caption for Figures 1, 2, 4, 5, 6 and 7 and clarified in the Methods (lines 434, 488-489, 559-561, 571-572, 579, 582, 588, 590, 607, 616-618, and 621-622).

2. Description of some mouse experiments in the Materials and methods section states that 4T1 cells are transplanted into mammary fat pad (orthotopic model). Does that apply to every experiment?

Response: In the manuscript, both subcutaneous and orthotopic tumor models were used. The experiment in Figure 5 used orthotopic 4T1 tumors in BALB/c mice to get a clinically relevant measurement of tumor clearance. Details about the location of tumors has been clarified in the captions for Figures 1, 2, 4, 5, 6 and 7 and the Methods (lines 366-374, 434, 488, 559, 607, and 616-617).

3. Were bacteria injected into the mice always through the tail vein?

Response: Bacteria were administered with both intratumoral and intravenous injections. Intravenous injections were used when bacterial tumor-targeting was a critical component of the experiment. This information has been clarified in the captions for Figures 1, 2, 4, 5, 6 and 7 and the Methods (lines 435, 489, 561, 571, 579, 582, 588, 590, 608, 618, 622).

4. The number of animals used per treatment should be clearly written in each figure.

Response: As requested, the number of mice has been added to the caption of each figure.

5. Labeling should be constant through the manuscript. For example: in the Fig2., it is written PsseJ-LysE, but in the Legend of the Fig2. It is written PsseJ-lysE.

Response: As suggested, all instances were written as *PsseJ-LysE*.

6. Adding color code in the figure, not only Figure legend, would make it easier for readers.

Response: As requested, color codes have been added to the figures.

Specific comments:

1. Fig4E.- How was bacteria delivered into the mice? What about expression of GFP in the lungs (Fig.4F.)?

Response: In this figure (4E in the original manuscript and 2H in the revision), ID Salmonella were administered by intratumoral injection. In this experiment, expression in the lungs was not measured. When we measured accumulation in the lungs, we found that the bacteria had cleared by 7 days (Figure S3B). To address this concern, text was added to the caption for Figure 2.

2. Is the drug delivery to 20% of tumor cells significant (Fig4I)? How long after bacterial injection tumors were harvested?

Response: The tumors in this figure (4I in the original manuscript and 4F in the revision), were harvested 4 days (96 h) after injection. In the transition zone, delivery of GFP with *flhDC*-expressing Salmonella to 21% of cells was significantly greater than zero ($P < 0.001$) and significant compared to controls ($P < 0.001$). To clarify this information, text was added to the caption for Figure 4F and the Results (lines 188-189).

3. Fig5C., some bacterial infiltration has been observed in the liver. Did the authors look for histological signs of liver inflammation/infiltration of immune cells? How long after bacterial injection was blood liver panel done (Fig5E)?

Response: For the results in Figure 5, the biodistribution (Figure 5C) and the comprehensive hematology and chemistry profiling (Figures 5D&E) were performed 14 days after bacterial injection. We have added additional results where the biodistribution was quantified earlier at 6 hours and 7 days (Figure S3), and the chemistry profile was quantified at 7 days (Figure S4). Inflammation was not measured in the liver, but at none of these times did we detect liver toxicity or an immune response (Figures 5D-E, S3, and S4). To address this concern, new results were added to the Supplemental Results (Figures S3 and S4) and text was added to the caption for Figure 5.

4. Was the experiment with NIPP1-CD (Figure 5) performed in 4T1 cells as well (in vitro and in vivo)?

Response: The efficacy of NIPP1-CD ID Salmonella was measured in 4T1 tumors. No difference was seen compared to saline controls. The results of this experiment were added to the Supplemental information (Figure S2). We included this data to show that ID Salmonella is capable of delivering multiple types of therapeutic proteins, and that proteins that are effective in vitro can have different responses in vivo. To address this point, text was added to the Results (lines 213-214), the Discussion (lines 256-258 and 304-309), and the Supplemental Results (Figure S2).

5. In the Fig6D successful delivery of NIPP1-CD to tumor cells was observed. Was there any effect on tumor cells? Apoptosis? It would be informative if delivery to the tumor cells is quantified (percentage of tumor cells with delivered NIPP1-CD).

Response: As requested, we quantified the fraction of cells that received NIPP1-CD in the experiment in Figure 6D. Based on area of viable cells and the area of protein delivery, we determined that approximately $23 \pm 5\%$ of cells received NIPP1-CD. We did not see any effect of this delivery of NIPP1-CD on the volume of 4T1 tumors in BALB/c mice (Figure S2). To address this concern, we have added the results from this experiment to the Supplemental Results (Figure S2) and added text to the Results (lines 212-214) and the Methods (lines 606-613).

6. Figure 7E, what about metastasis?

Response: In the BALB/c mice in Figure 7E, metastases did not form. The 4T1 tumor model is very aggressive and often mice have to be sacrificed before the formation of metastases. Because of this aggressive growth, we formed lung metastases by intravenous injection of 4T1 cells. The effect of *CT Casp-3* Salmonella on these metastases is shown Figures 7F&G. Intravenous injection of these bacteria prevented metastases from growing much bigger than their initial volume. Many of the control metastases that were treated with Paclitaxel, the standard-of-care, grew exponentially by three weeks from injection. To address this concern, text was added to the caption for Figure 7.

7. Figure 7C, there is significant cell death observed even in the tumor masses treated only with control. Can authors comment on that?

Response: The addition of bacteria to cell masses in microfluidic device (as in Figure 7C) always causes some cell death. We have reported this phenomenon previously (Toley BJ and Forbes NS. 2012. *Integr. Biol.* 4:165). In Figure 7C, the difference in cell death between *CT Casp-3* Salmonella and controls is caused by the delivery of the exogenous protein. To address this concern, text was added to the caption for Figure 7 and the Results (lines 222-224).

8. Please clarify if in the Figure 7G, are metastasis formed in orthotopic model or by cells injected through tail vein?

Response: The metastases in Figure 7G were formed by injection of cells into the tail vein. This information has been clarified in the caption for Figure 7.

9. Was safety study (Figure 5) performed with Salmonella strains that release caspase3 and NIPP1-CD? Because some bacteria colonize liver and spleen, releasing these proteins could potentially induce damage.

Response: The reviewer is correct; the safety study in Figure 5 was performed with ID Salmonella that did not release *CT Casp-3*. To address this concern, we performed an additional safety study with *CT Casp-3* Salmonella (Figure S4). We observed that neither *CT Casp-3* Salmonella nor ID

Salmonella caused any observable toxicity in tumor-free mice. These new results were added to the Supplemental Results (Figure S4) and text was added to the Results (lines 228-231) and the Discussion (lines 314-320).

10. Was there any toxicity observed in mice treated long-term with bacteria?

Response: We did not observe any toxicity in mice that were treated long-term with bacteria. In healthy mice, bacteria were cleared from most organs by seven days (Figure 5 and S3). In these mice, no toxicity was observed. In tumor-bearing mice, most adverse events were caused by presence of cancer cells. In mice with eliminated or severely delayed tumors (> 50 days, Figure 7H), no adverse effects were seen from the bacterial treatment. To address this point, text has been added to the Results (lines 243-244).

REVIEWERS' COMMENTS

Reviewer #1 (Remarks to the Author):

Most of my concerns have been well addressed. The supplemented biodistribution shown in Figure S3 indicated significantly enhanced accumulation of CT Casp-3 than normal Salmonella in all the examined major organs. What were the main causes? Did this mean more bacteria invaded into normal tissue cells? If this was the case, why elevated accumulation of bacteria in these organs did not cause any side effects? These should be explained and discussed as the safety issue is the key element for the application of these engineered bacteria for treatment. Readers would also be benefited quite a lot with these clarifications.

Reviewer #2 (Remarks to the Author):

Authors answered and clarified most of my questions.

Reviewer #3 (Remarks to the Author):

Most of my concerns have been addressed. Minor changes should be added in the Figure Legends of Figure 4 and 7. The authors should describe the cells used in Figure 4B and add the number of mice used in Figure 7I.

Response to Reviewers

Intracellular delivery of protein drugs with an autonomously lysing bacterial system reduces tumor growth and metastases

Vishnu Raman^{*,1,2}, Nele Van Dessel^{*,1,2}, Christopher L. Hall^{1,2}, Victoria E. Wetherby², Samantha A. Whitney¹, Emily L. Kolewe¹, Shoshana M.K. Bloom¹, Abhinav Sharma¹, Jeanne A. Hardy^{3,4,5}, Mathieu Bollen⁶, Aleyde Van Eynde⁶, Neil S. Forbes^{1,2,4,5,†}

Submitted for publication in *Nature Communications*

Manuscript ID: NCOMMS-20-34783B

We would like to thank the editors and reviewers for their thoughtful comments, which have all been addressed. Many of the responses and modifications incorporated in the revision are listed below. Text that was altered from the original manuscript is indicated in red. This document includes responses to the two comments of the reviewers.

Response to Reviewer's Comments

Reviewer #1

Most of my concerns have been well addressed. The supplemented biodistribution shown in Figure S3 indicated significantly enhanced accumulation of CT Casp-3 than normal Salmonella in all the examined major organs. What were the main causes? Did this mean more bacteria invaded into normal tissue cells? If this was the case, why elevated accumulation of bacteria in these organs did not cause any side effects? These should be explained and discussed as the safety issue is the key element for the application of these engineered bacteria for treatment. Readers would also be benefited quite a lot with these clarifications.

Response: The reviewer raises an important concern. We often see Salmonella in major organs at six hours after injection (as seen in Supplementary Figure 2a). This is caused by the bacteria present in the blood in these organs. These bacteria do not interact with the tissues and do not cause adverse effects (as shown in Figure 5c-e and Supplementary Figures 1 and 4). Over time the immune system clears the bacteria from the blood and organs, as can be seen in Figure 5c and Supplementary Figure 3b. To address this concern, we added text to the Results on lines 196-198, the Discussion on lines 316-319, and the caption of Supplementary Figure 2.

Reviewer #2

Authors answered and clarified most of my questions.

Reviewer #3

Most of my concerns have been addressed. Minor changes should be added in the Figure Legends of Figure 4 and 7. The authors should describe the cells used in Figure 4B and add the number of mice used in Figure 7I.

Response: As suggested by the reviewer, the name of the cell line was added to legend of Figure 4b and the number of mice has been added to the legend of Figure 7i.